# TRIM2 E3 ligase substrate discovery reveals zinc-mediated regulation of TMEM106B in the endolysosomal pathway

Cecilia Perez-Borrajero [1✉], Frank Stein [2], Kristian Schweimer [3], Mandy Rettel [2], Jennifer J Schwarz [2], Per Haberkant [2], Karine Lapouge [4], Jesse Gayk [1], Thomas Hoffmann [1], Sagar Bhogaraju [5], Kyung-Min Noh [6], Mikhail Savitski [1,2], Julia Mahamid [1,7] & Janosch Hennig [1,3✉]

## Abstract

**TRIM2 is a mammalian E3 ligase with particularly high expression in Purkinje neurons, where it contributes to neuronal development and homeostasis. The understanding of ubiquitin E3 ligase function hinges on thoroughly identifying their cellular targets, but the transient nature of signaling complexes leading to ubiquitination poses a significant challenge for detailed mechanistic studies. Here, we tailored a recently developed ubiquitin-specific proximity labeling tool to identify substrates of TRIM2 in cells. We show that TRIM2 targets proteins involved in the endolysosomal pathway. Specifically, we demonstrate using biochemical and structural studies, that TRIM2 ubiquitinates TMEM106B at lysine residues located in the cytosolic N-terminal region. Substrate recognition involves a direct interaction between TRIM2 and a newly identified zinc-coordination motif in TMEM106B that mediates homo-dimerization, is required for specific protein–protein interactions, and lysosomal size regulation. We found that in addition to catalysis, the tripartite motif is involved in substrate recruitment. Our study thus contributes a catalog of TRIM2 effectors and identifies a previously unrecognized regulatory region of TMEM106B crucial to its function.**

**Keywords** TRIM2; Ubiquitination; TMEM106B; Lysosome; Zinc-binding
**Subject Categories** Membranes & Trafficking; Post-translational Modifications & Proteolysis; Structural Biology

## Introduction

The post-translational modification (PTM) of proteins with ubiquitin is one of the major cellular signals that maintain proteome homeostasis ("proteostasis") (Bett, 2016; Franić et al, 2021; Kaushik and Cuervo, 2015). Imbalances in the levels, localizations, and folding states of a multitude of protein substrates can be countered by the concerted action of E1 activating enzymes, E2 conjugating enzymes, and E3 ligases in an ATP-dependent manner (Pickart, 2001; Swatek and Komander, 2016). As a result of this enzymatic cascade, the C-terminal glycine residue of ubiquitin is covalently linked to a particular target, most commonly through a lysine-mediated isopeptide bond (Pickart, 2001; Swatek and Komander, 2016). Mono- or poly-ubiquitin modification leads to various downstream effects, including protein trafficking and degradation through recruitment of ubiquitin binding entities that can specifically recognize the "ubiquitin code" (Komander and Rape, 2012; Pickart, 2001; Swatek and Komander, 2016).

The loss of proteostasis is a hallmark of diseases related to aging (López-Otín et al, 2023). Many of the genes associated with neurodegeneration are involved in protein quality control, with E3 ligases being a particularly prominent group among these (Gan et al, 2018; Liu et al, 2023). Because these enzymes catalyze the last and most specific step in the ubiquitination cascade, the identification of their substrates has been the focus of extensive work to understand their biological roles (Iconomou and Saunders, 2016). However, substrate-ligase interactions are weak and transient, making the systematic identification of these pairings challenging (Iconomou and Saunders, 2016). Recently, versatile proximity-based labeling methods specific for enriching ubiquitinated proteins have been developed (Barroso-Gomila et al, 2023; Huang et al, 2024; Mukhopadhyay et al, 2024). While still relying on the physical association of the ligase and the substrate, these tools additionally require ubiquitin to be covalently attached, and

[1]Molecular Systems Biology Unit, European Molecular Biology Laboratory, Heidelberg, Germany. [2]Proteomics Core Facility, European Molecular Biology Laboratory, Heidelberg, Germany. [3]Chair of Biochemistry IV, Biophysical Chemistry, University of Bayreuth, Bayreuth, Germany. [4]Protein Expression and Purification Core Facility, European Molecular Biology Laboratory, Heidelberg, Germany. [5]Structural Biology Unit, European Molecular Biology Laboratory, Grenoble, France. [6]Genome Biology Unit, European Molecular Biology Laboratory, Heidelberg, Germany. [7]Cell Biology and Biophysics Unit, European Molecular Biology Laboratory, Heidelberg, Germany. ✉E-mail: cecilia.perez@embl.de; janosch.hennig@uni-bayreuth.de

thereby address the difficulties of identifying the short-lived interactions that are commonplace in signaling cascades.

The E3 ligase TRIM2 belongs to the tripartite motif (TRIM) family, defined by the presence of an N-terminal RING domain, one or two B-boxes, and a coiled-coil domain that mediates dimerization (Williams et al, 2019). The C-terminal regions in the TRIM family are thought to recruit substrates and are thus associated with target specificity (Wang and Hur, 2021; Williams et al, 2019). TRIM2 contains Filamin and NHL (NCL-1, HT2A and Lin-41 homology) domains and thus belongs to the TRIM-NHL subfamily (Williams et al, 2019). Animal models of TRIM2 deletion develop uncontrolled muscle movements (i.e., ataxia) in ageing mice, and peripheral neuropathy associated with the loss of Purkinje neurons of the cerebral cortex, where TRIM2 is particularly abundant (Li et al, 2020). However, this phenotype is not strictly dependent on the ubiquitination activity of TRIM2. The neuromuscular disorder Charcot-Marie-Tooth Disease is associated with TRIM2 mutations in humans and a dysregulation of endosomal trafficking in cell culture models (BasuRay et al, 2013; Lee et al, 2012; Ylikallio et al, 2013). These and many other observations point to TRIM2 having crucial roles in neuronal health and differentiation (Balastik et al, 2008; Khazaei et al, 2011; Lokapally et al, 2020).

At the molecular level, TRIM2 has been shown to ubiquitinate neurofilament light chain (NEFL) and BCL-2-interacting mediator of cell death (BIM) to influence axon stability and apoptosis, respectively (Balastik et al, 2008; Thompson et al, 2011). However, the catalog of substrates of TRIM2 and associated cofactors is still largely unknown, and there is little mechanistic insight into how ubiquitination occurs, including its linkage preferences. In this study, we tailored proximity-based labeling tools coupled to quantitative mass spectrometry (MS) to uncover previously unknown ubiquitination substrates and cofactors associated with TRIM2 activity. We identified the lysosomal protein TMEM106B using this approach and validated that it is a bona fide TRIM2 target. Specifically, we show that TRIM2 ubiquitinates the N-terminal cytosolic region of TMEM106B in vitro, and that the two proteins interact directly. TMEM106B was previously shown to exhibit molecular weights consistent with dimerization as determined by SDS-PAGE (Chen-Plotkin et al, 2012; Held et al, 2025). We find here that substrate recruitment by TRIM2 occurs through the N-terminal cytosolic region of TMEM106B involved in homodimerization. Dimerization is mediated by binding of a zinc metal ion through two specific cysteine residues in each monomer. We determined the structure of this motif using NMR spectroscopy, and we demonstrate that zinc binding is key to the formation of specific protein–protein interactions and the function of TMEM106B in lysosomal size regulation.

## Results and discussion

### Identification of TRIM2 ubiquitination targets in the endolysosomal pathway

We used a ubiquitin-specific proximity labeling approach described recently (Mukhopadhyay et al, 2024) and adapted it for the identification of ubiquitination substrates of TRIM2 (Fig. 1A). In this method, an optimized AviTag present on ubiquitin (AviTag-

Ub) allows specific BirA-mediated biotinylation of post-translationally modified substrates upon recruitment by BirA-tagged TRIM2. Although TRIM2 has prominent roles in the brain, neurons do not readily take up exogenous DNA (Karra and Dahm, 2010). HEK 293-T cells were used to allow for efficient co-transfection and protein expression of the AviTag-Ub and BirA-TRIM2 plasmids simultaneously (Fig. 1B; Appendix Fig. S1).

We found that WT TRIM2 was expressed at low levels in the absence of proteasome inhibition, and protein levels decreased further upon co-transfection with AviTag-Ub (Fig. 1B). However, single point mutations of TRIM2 targeting the coordination of structural zinc atoms on the RING domain (C23S or C60S) led to its accumulation in cells without the need for proteasome inhibition (Fig. 1B). Autoubiquitination of TRIM2 in cells has been previously reported (Balastik et al, 2008), but it was unclear whether this led to its degradation. Our results suggest that TRIM2 can be degraded in a proteasome-dependent manner aided by its own catalytic activity. Based on these results, we carried out the experiments in the presence of proteasome inhibition. Protein substrates were identified using MS based on tandem mass tag (TMT) labeling (Dayon et al, 2008) after enrichment of the biotinylated substrates with streptavidin pulldowns (Fig. 1A).

We identified 29 proteins enriched in the presence of the WT variant relative to both negative controls (BirA and BirA-TRIM2[dead] (C60S)) (Fig. 1C, Dataset EV1). None of these had been previously reported as TRIM2 substrates, consistent with the capture of transient interactions possible with the use of proximity labeling methods. Enrichment analyses found gene ontology (GO) terms such as ubiquitin-like protein ligase binding (Appendix Fig. S1), consistent with its E3 ligase role. Notably, we found terms related to multivesicular body transport, endocytosis, and autophagy (Fig. 1D), due to the presence of proteins such as HGS, STAM, STAM2, and RABEP2 (Fig. 1C). Genes related to the antiviral RIG-I-like receptor pathway also featured prominently (e.g., TRIM25, TRAF3, CYLD) (Fig. 1D and Dataset EV1). Despite conducting our experiments in HEK 293-T cells, we also found several proteins associated with neuronal homeostasis, such as TMEM106B, ITM2B, YARS, SLC20A1, and TRIB3 (Fig. 1C). Thus, TRIM2 can associate with a wide variety of cytosolic and membrane proteins. Its link to multivesicular bodies and the ESCRT-0 components STAM and STAM2 suggest that ubiquitination of substrates destined for the endolysosomal pathway is its primary function as a ubiquitin ligase.

### TRIM2 ubiquitinates the N-terminus of TMEM106B in vitro

From the substrates identified through proximity labeling, we focused on TMEM106B, which is highly expressed in Purkinje cells and has links to motor neuron degeneration and the endolysosomal pathway similarly to TRIM2 (Chang et al, 2022; Feng et al, 2022; Rademakers et al, 2021; Schweighauser et al, 2022; Schwenk et al, 2014; Stroobants et al, 2021). TMEM106B contains a putatively disordered N-terminal cytosolic region and a globular C-terminal domain projecting either into the lysosomal lumen or the cellular periphery, where it has been reported to act as an alternative receptor for SARS-CoV-2 (Baggen et al, 2021, 2023; Chang et al, 2022; Kang et al, 2018; Schweighauser et al, 2022) (Fig. 2A). We established in vitro ubiquitination assays to examine TRIM2

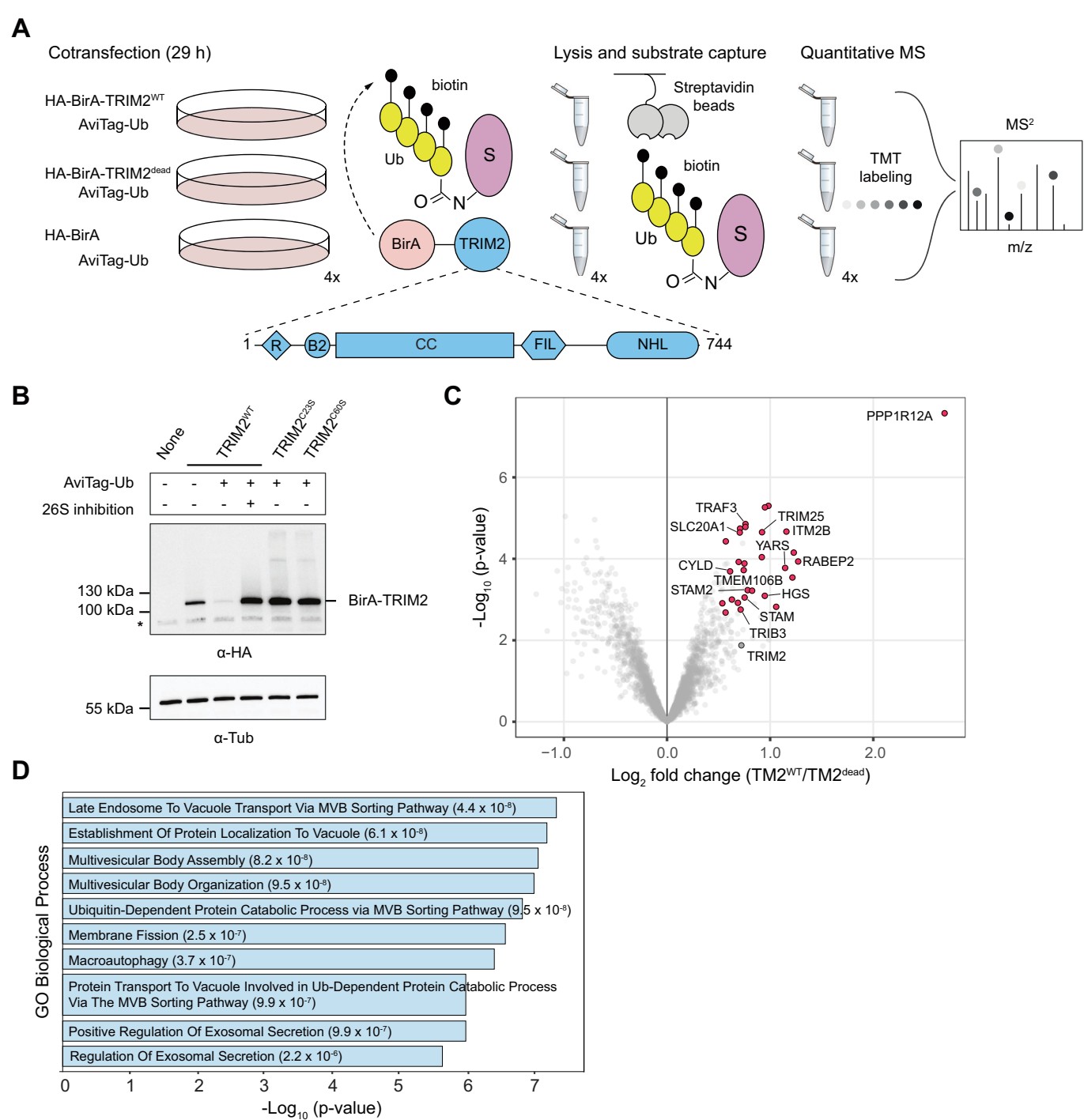

**Figure 1. Identification of ubiquitination substrates of TRIM2 in HEK 293-T cells.**

(**A**) Experimental workflow used in the enrichment of TRIM2 ubiquitination targets. Adherent HEK 293-T cells were co-transfected with AviTag-Ub along with either HA-BirA-TRIM2<sup>WT</sup>, a catalytically dead variant (HA-BirA-TRIM2<sup>C60S</sup>), or HA-BirA. Protein enrichment and identification was accomplished through streptavidin-based pulldowns followed by TMT labeling prior to MS from two biological and two technical replicates (n = 4). The domain organization of TRIM2 with the relative positioning of known structured domains is indicated below. The following abbreviations are used for the structured domains: R for RING, B for B-box, CC for coiled-coil, and FIL for Filamin. (**B**) The protein expression level was assessed using immunoblotting against the HA tag (above) or tubulin controls (below) in transfected HEK 293-T cells. These were replicated at least three times in the laboratory. (**C**) Identified TRIM2 ubiquitination substrates and/or associated proteins, highlighted in red if they were enriched in the WT condition relative to both negative controls and not significantly different between these. The GO enrichment analysis of those is shown in (**D**), to highlight biological process terms (see also the Appendix Fig. S1). The p-values reported were determined using limma's empirical Bayes-moderated t-test or Fisher's exact/ hypergeometric tests in the case of MS and enrichment analyses, respectively. Source data are available online for this figure.

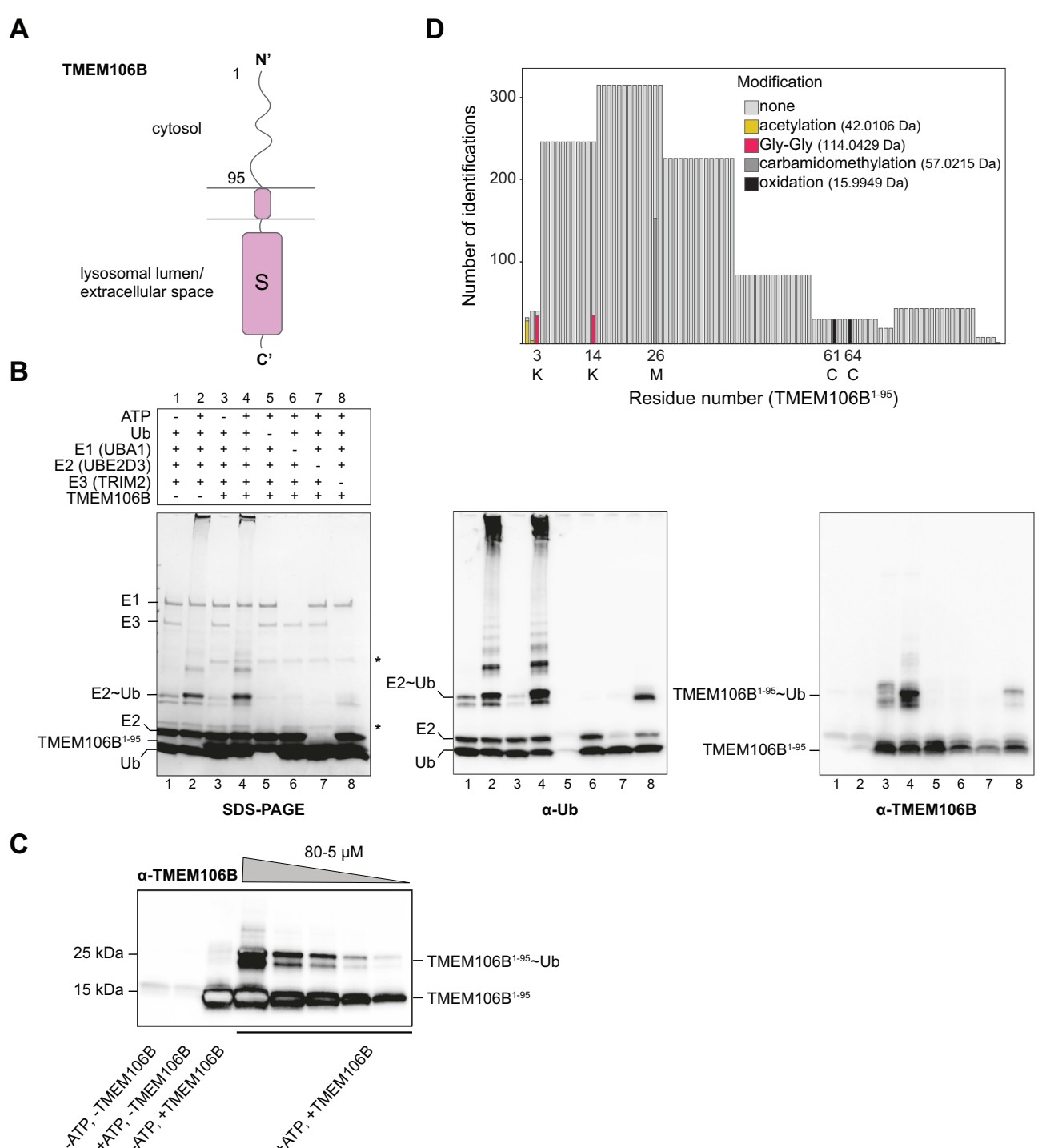

**Figure 2. The cytosolic region of TMEM106B is ubiquitinated by TRIM2.**

(A) Domain organization and corresponding cellular location of TMEM106B depicted diagrammatically. (B) In vitro ubiquitination assays involving the enzymatic cascade components and the N-terminal cytosolic region of TMEM106B (1–95). Minor impurities from protein degradation during purification are marked with an asterisk. The samples produced were assessed through Coomassie stained SDS-PAGE (left), immunoblots against ubiquitin (middle), and TMEM106B (right). Note that the antibody against ubiquitin cross-reacts with the E2 (compare lanes 6 and 7). (C) Similar ubiquitination assays with varying TMEM106B$^{1-95}$ concentration, from 80 to 5 μM (final). The presence of two high MW bands upon addition of ATP shows that both mono- and di-ubiquitination can occur. (D) The region corresponding to ubiquitinated TMEM106B was excised from the polyacrylamide gel and analyzed for PTMs with MS. Shown are the modification sites identified along the TMEM106B protein sequence, with the amino acids involved indicated below using one-letter codes. Note that save for the Gly-Gly modification, all other PTMs occur due to sample preparation prior to MS. These results could be replicated at least three times in the laboratory. Source data are available online for this figure.

enzymatic activity in a minimal system using purified components from recombinant expression in *E. coli* and monitored ubiquitination upon addition of ATP with SDS-PAGE or immunoblotting (Fig. 2B; Appendix Fig. S2). We confirmed that TRIM2 is autoubiquitinated and identified the modification sites to occur mostly in the disordered linker (~50 amino acids) separating the FIL and NHL domains, with few modifications also occurring in the coiled-coil region (Dataset EV2). Given that TRIM2 is cytosolic (Khazaei et al, 2011; Xiao et al, 2022), we then tested ubiquitination of the N-terminus of TMEM106B (residues 1–95) (Fig. 2B). Parallel to autoubiquitination of TRIM2, we observed the formation of species corresponding to mono- and di-ubiquitinated TMEM106B$^{1-95}$ over a range of substrate concentrations (Fig. 2B,C). Thus, TRIM2 catalyzes the modification of TMEM106B using the classical ubiquitin cascade components without the need for additional cellular cofactors.

The region of TMEM106B used in these assays contains three lysine residues at positions 3, 14, and 95, which could serve as potential acceptors for the modification. Using MS, we could assign both lysine 3 and 14 as ubiquitination sites based on the presence of the signature Gly-Gly dipeptide mass increase of ~114 Da that results after trypsin digestion (Fig. 2D, Dataset EV2).

A ubiquitin chain composed of four moieties is minimally required to promote proteasome-dependent substrate degradation (Thrower et al, 2000). However, additional cellular factors absent in the above assay could extend ubiquitinated TMEM106B to promote its turnover. We tested whether TRIM2 had a role in TMEM106B degradation through the proteasome to clarify the role of its ubiquitination. Transient co-transfections of TMEM106B and TRIM2 in HEK 293-T cells showed that the presence of WT TRIM2 did not alter TMEM106B expression relative to catalytically dead TRIM2 variants, nor did proteasome inhibition stabilize its levels (Appendix Fig. S2). This is in line with previous observations that TMEM106B is not turned over through the proteasome (Brady et al, 2013) and suggests that ubiquitination of TMEM106B by TRIM2 has an alternative role.

## TRIM2 interacts directly with a zinc-coordinated homodimerization interface of TMEM106B

The C-terminal domains of TRIM proteins are thought to be involved in substrate recruitment, but this has only been demonstrated for a few PRYSPRY-containing members (Wang and Hur, 2021). We set out to investigate the interaction between TRIM2 and TMEM106B using NMR spectroscopy. The $^1$H-$^{15}$N HSQC spectrum of TMEM106B$^{1-95}$ indicated that the region is mostly disordered as previously shown (Kang et al, 2018). However, a few residues exhibited spectral dispersion consistent with some degree of ordering, which had not been described (Fig. 3A). Upon addition of full-length TRIM2 to $^{15}$N-labeled TMEM106B$^{1-95}$, we observed line broadening (i.e., signal loss) in several amide peaks, as would be expected if an interaction took place resulting in the slower tumbling of TMEM106B in solution (Fig. 3A,B). Notably, the largest changes in linewidth occurred in the most dispersed amide signals, indicating that TRIM2 recognizes a structured motif. To determine which domains were responsible for the interaction, we repeated this experiment with either the N-terminal RING, B-box, and coiled-coil region of TRIM2 (RBCC, residues 8–318), or the C-terminal FIL and NHL domains (residues 318–744). We

found that both the RBCC and FIL-NHL domains caused line broadening and therefore contributed to binding (Fig. 3B; Appendix Fig. S3).

To identify the interaction interface, we assigned the backbone chemical shifts of TMEM106B$^{1-95}$ and quantified the signal loss in $^1$H-$^{15}$N-HSQC spectra as a function of residue number. The largest perturbations were centered around residue 60 in the C-terminal region of TMEM106B, which showed consistent changes in the presence of full-length TRIM2, as well as the RBCC and FIL-NHL regions (Fig. 3B). The AlphaFold2 model of TMEM106B predicts that cysteine residues 61 and 64 in this region could form a disulfide bond, given that the two sulfhydryl groups are in very close proximity (<5 Å) and the peptide backbone forms a hairpin around them (Appendix Fig. S3) (Jumper et al, 2021). This observation prompted us to test if the oxidation state of TMEM106B was important to its structure and recognition of TRIM2. However, upon addition of 10 mM of dithiothreitol (DTT, a reducing agent) to $^{15}$N-labeled TMEM106B$^{1-95}$, we observed no significant changes in the spectra, indicating that highly reducing conditions do not change its conformation (Appendix Fig. S3).

Previous findings indicated that TMEM106B could oligomerize, but it was unclear how and to what extent (Brady et al, 2013). The predicted MW of monomeric TMEM106B$^{1-95}$ is 10.7 kDa. Using size exclusion chromatography coupled to multi-angle light scattering (SEC-MALS), we determined that at concentrations as low as ~0.3 mg/mL, the protein had a MW of 23.1 ± 0.4 kDa and is therefore dimeric (Fig. 3C). We used the AlphaFold2-Multimer pipeline to predict the structure of homodimeric TMEM106B$^{1-95}$ (preprint: Evans et al, 2022). The prediction suggested that residues C61 and C64 could participate in metal coordination, as the side chains arising from each of the two monomers formed a tetrahedral arrangement commonly observed in metal binding sites (Fig. 3D; Appendix Fig. S3). In addition, the chemical shifts of the $^{13}$C$^\beta$ atoms of C61 and C64 were ~32 ppm, which is characteristic of cysteine residues participating in zinc-coordination (Appendix Fig. S3) (Kornhaber et al, 2006; Martin et al, 2010).

To test whether metal coordination was important in promoting dimerization, we compared the $^1$H-$^{15}$N HSQC spectra of TMEM106B$^{1-95}$ in the absence and presence of excess ethylenediaminetetraacetic acid (EDTA, a metal chelator). Since metal ions can bind with very high affinity to protein coordination sites (Kluska et al, 2018), we increased the temperature to 45 °C to promote metal ion release. Indeed, we observed that the presence of EDTA and heat treatment caused new amide peaks to appear in the center of the spectrum, consistent with a monomeric form of TMEM106B (Fig. 3E). In the absence of EDTA, the protein did not exhibit changes in the spectrum after heating in the same manner, and thus the changes observed are not due to thermal denaturation of the dimer (Fig. 3E). In addition, SEC-MALS experiments detected a molecule with a MW of ~14 kDa only in the presence of EDTA and heat treatment, indicative of a shift in the equilibrium to the monomeric state (Appendix Fig. S3). Next, we introduced the point mutations C61S and C64S to prevent metal coordination. This variant was monomeric with a MW of 11.6 ± 0.2 kDa (Fig. 3C).

Given the large number of zinc-binding proteins that exist in the human proteome, estimated to be ~10%, and the prominent role of zinc in neuronal homeostasis, we hypothesized that the metal ion mediating TMEM106B dimerization could be zinc (Andreini et al, 2006; Frederickson et al, 2000). We thus used a sample with a

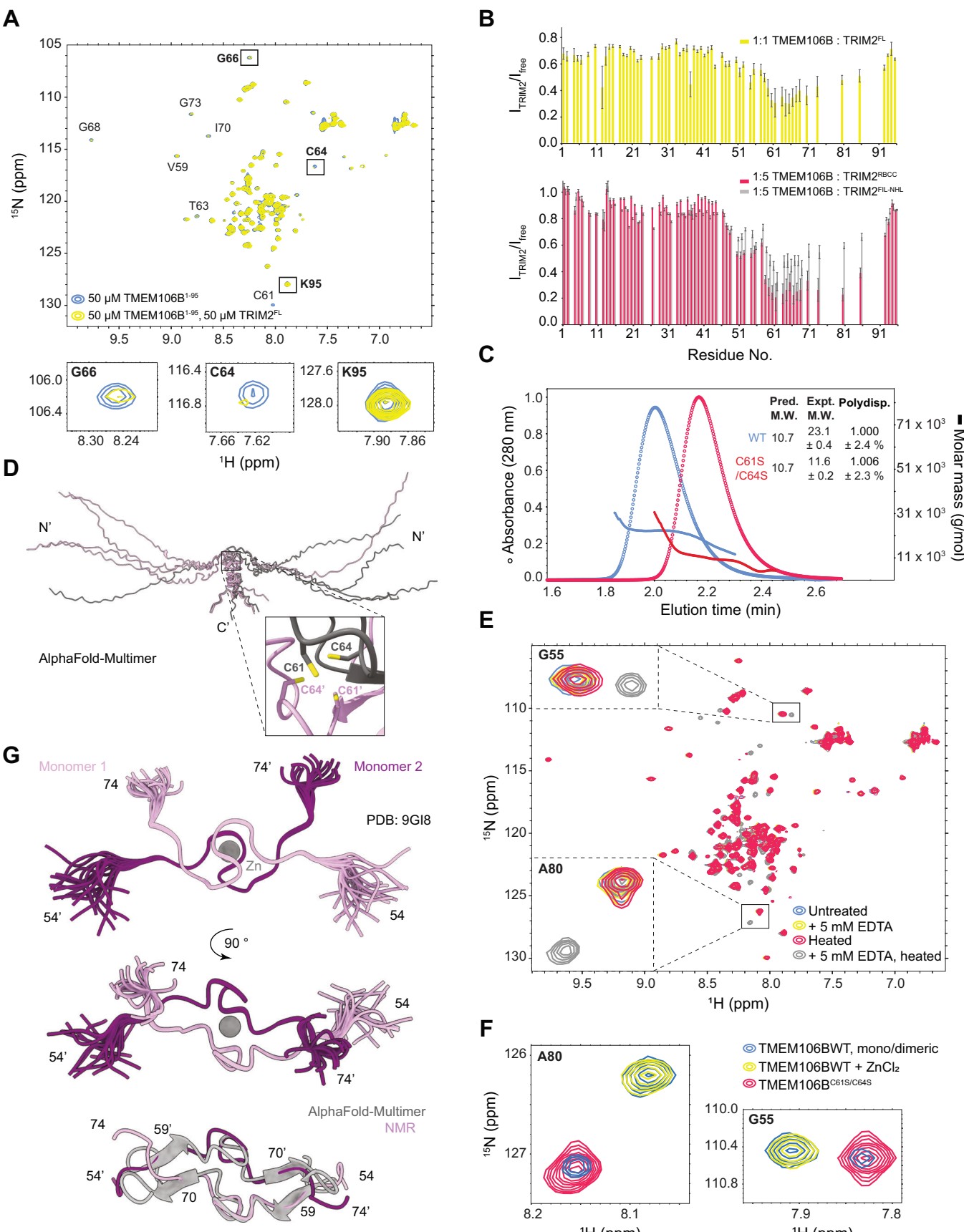

**Figure 3.   TRIM2 interacts directly with a homodimerization interface present in TMEM106B driven by zinc binding.**

(A) The $^1$H-$^{15}$N HSQC of $^{15}$N-labeled WT TMEM106B$^{1-95}$ was recorded in the absence and presence of equimolar amounts of full-length TRIM2. Amide peaks exhibiting particularly large changes are labeled, along with K95 serving as a negative control. Shown below are zoomed-in views of illustrative examples. The signal loss per residue assigned, computed as the ratio of peak heights in the presence and absence of TRIM2 variants is shown in (B). The error bars correspond to the cumulative error arising from the two spectra used for quantification, obtained from signal-to-noise ratio estimates using NMRFAM-Sparky. (C) SEC-MALS experiments comparing WT and C61S/C64S mutant TMEM106B$^{1-95}$. The theoretical MW, determined from the protein sequence and assuming a monomer, is shown next to the experimentally determined values (units of kDa). (D) AlphaFold2-Multimer predictions of the top 5 ranked structural models of homodimeric TMEM106B$^{1-95}$ after alignment of the main chain. Shown in a zoomed view for one model below, are cysteines 61 and 64 that form a tetrahedral arrangement. The position errors are shown in Appendix Fig. S3. (E, F) Overlaid $^1$H-$^{15}$N HSQC spectra of WT TMEM106B$^{1-95}$ in the presence and absence of 5 mM EDTA, with or without heat treatment (45 °C, 2 h). Peaks consistent with a monomeric species are observed only upon treatment with both EDTA and heat. The inset shows zoomed views of two regions corresponding to residues glycine 55 and alanine 80 as examples. (F) Similarly overlaid spectra of $^{15}$N-labeled WT TMEM106B$^{1-95}$ exhibiting mixed monomeric and dimeric populations before and after supplementation with 100 μM ZnCl$_2$, along with the monomeric C61S/C64S mutant. (G) NMR-derived structural ensemble of dimeric TMEM106B$^{54-92}$ (top and middle, PDB: 9GI8). Shown is the alignment of the 20 lowest energy models including residues 54–74 that are located around the zinc binding site. Bottom: overlay of the top ranked models obtained from NMR or AlphaFold2-Multimer for the same region. The zinc atom has been removed for clarity. Biophysical measurements were performed once or twice in the laboratory. Source data are available online for this figure.

mixture of monomeric and dimeric forms of TMEM106B and compared the $^1$H-$^{15}$N HSQC upon supplementation with ZnCl$_2$. Indeed, we found that the amide peaks corresponding to monomeric TMEM106B disappeared, while the intensity of peaks corresponding to the dimeric species increased upon addition of the metal ion (Fig. 3F). Therefore, the association of TMEM106B with zinc determines the monomer-dimer equilibrium observed in solution.

The accuracy of structural predictions is currently well established for ordered (i.e., folded) protein domains (preprint: Evans et al, 2022; Jumper et al, 2021). However, flexible regions of proteins are not often characterized experimentally and exhibit poor prediction scores by the state-of-the-art tools available (Terwilliger et al, 2023). We determined the NMR-derived structure of dimeric TMEM106B and compared it to the AlphaFold2-Multimer predictions obtained above (Fig. 3G; Appendix Fig. S3, Appendix Table S1, PDB: 9GI8). To do this, we used a synthetic peptide spanning residues 54 to 92 supplemented with Zn$^{2+}$, corresponding to the most ordered residues that recapitulated the conformation observed in the longer TMEM106B$^{1-95}$ construct (Appendix Fig. S3). The structure shows that residues 57–71 directly involved in zinc coordination form a structure that is highly consistent with the AlphaFold2-Multimer model (Fig. 3G). Superposition of the highest ranking experimentally derived and predicted models resulted in a main-chain RMSD of 1.2 Å for those residues. Note that residues 69–71 and 58–60 adopt small β-strands visible only in some of the predicted models. Around three quarters of the predicted models contain additional β-strands involving residues 80–83 that appear to further stabilize the dimer but are not experimentally validated. The NMR data shows that the remainder of TMEM106B (residues ~72–92) does not adopt stable secondary structure, exhibiting random coil chemical shifts (see BMRB deposition 52589). Altogether, our data show that both the N- and C-terminal domains of TRIM2 interact directly with the cytosolic region of TMEM106B through a newly identified zinc-dependent homodimerization interface involving cysteine coordinating residues C61 and C64 contributed by each monomer.

We next tested whether zinc-mediated dimerization had an impact on the interaction with TRIM2. Monomeric TMEM106B$^{C61S/C64S}$ lacked the dispersed amide peaks observed in the WT variant, consistent with the loss of ordering of residues

54–74 (compare Figs. 3A and 4B). Upon addition of TRIM2$^{RBCC}$ to TMEM106B$^{C61S/C64S}$, very few amide peaks exhibited line broadening or chemical shift perturbations, and thus the interaction with TRIM2 is disrupted upon removal of the metal binding site (Fig. 4A,B). Specifically, the average ratio of signal intensities in the presence of TRIM2$^{RBCC}$ relative to free TMEM106B$^{1-95}$ ($I_{RBCC}/I_0$) for well-resolved amide peaks was 0.67 ± 0.28 and 0.82 ± 0.15 in the case of WT and monomeric variants, respectively (Fig. 4B). Despite impaired TRIM2 association, the monomeric variant could be similarly ubiquitinated by full-length TRIM2 (Fig. 4C), indicating that TMEM106B dimerization is not required for ubiquitination in vitro, although it may have an impact in the cellular context.

## Zinc binding by TMEM106B is important in protein–protein interactions and lysosomal regulation

TMEM106B regulates lysosomal acidification, size, and motility along microtubules (Stagi et al, 2014). Notably, increased levels of TMEM106B result in enlarged and functionally defective lysosomes (Brady et al, 2013; Busch et al, 2016). We leveraged this phenotype to understand the role that ubiquitination and zinc-mediated dimerization might have, using live confocal imaging of HEK 293-T cells transfected with variants of full-length TMEM106B (Fig. 5A). The expression of the mutants was comparable to WT, although the lysine deficient variant (TMEM106B$^{K3A/K14A}$) exhibited slightly lower protein levels (Appendix Fig. S4). Overexpression of TMEM106B$^{WT}$ caused an increase in lysosomal size as expected, in addition to perturbations in the intraluminal pH, as judged by the heterogeneous distribution of signal intensities arising from the pH-sensitive dye (LysoRed) (Fig. 5A). The phenotype from the lysine deficient mutant (TMEM106B$^{K3A/K14A}$) was not distinct from WT. However, cells transfected with TMEM106B$^{C61S/C64S}$ resembled the negative vector control, demonstrating that monomeric TMEM106B is not functional (Fig. 5A). Using immunostaining, we confirmed the membrane localization of the overexpressed TMEM106B variants, which did not show obvious differences (Appendix Fig. S4).

Next, we investigated whether the TMEM106B variants made distinct protein–protein interactions that could help explain the phenotype observed above. We performed co-immunoprecipitation followed by quantitative MS using transiently transfected HEK 293-

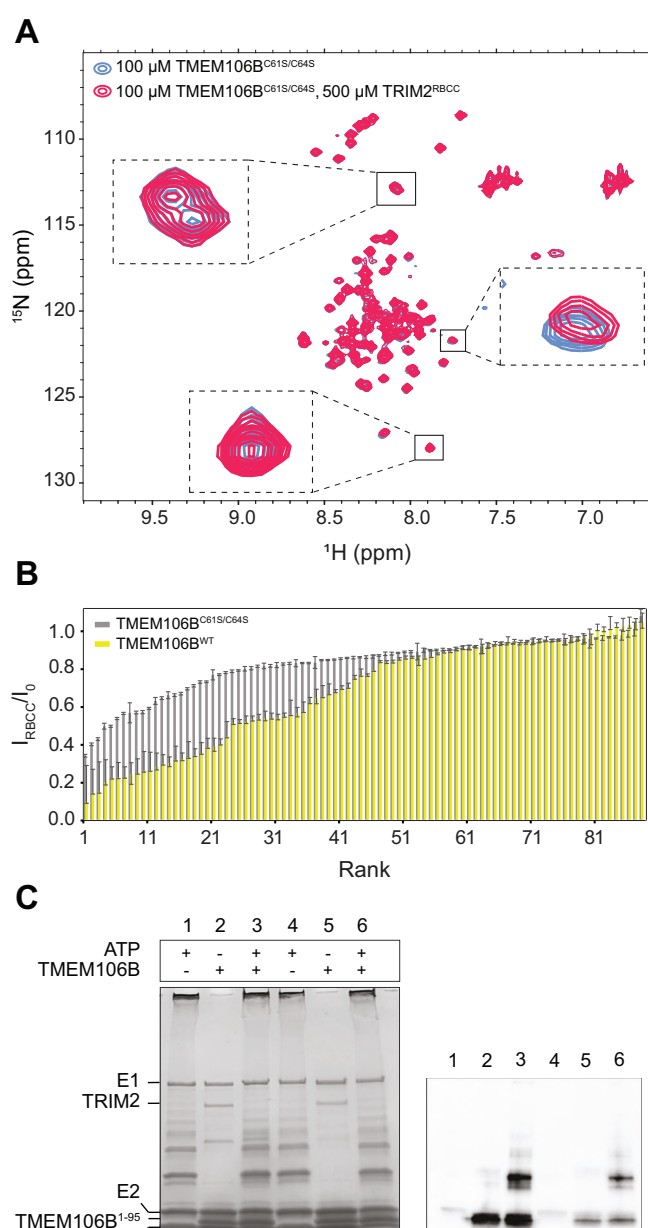

**Figure 4. Zinc binding is important in the association of TMEM106B with TRIM2.**

(A) Overlaid $^1$H-$^{15}$N HSQC spectra of $^{15}$N-labeled TMEM106B$^{C61S/C64S}$ in the absence and presence of 5-fold molar TRIM2$^{RBCC}$. Shown as insets are zoomed-in views of three sample amide peaks. (B) Line broadening upon addition of 5-fold molar excess TRIM2$^{RBCC}$ was quantified as the ratio of peak heights in the presence and absence of TRIM2$^{RBCC}$ ($I_{RBCC}/I_0$), for TMEM106B$^{WT}$ and TMEM106B$^{C61S/C64S}$ (see also Fig. 3A). The ratios are plotted as a function of the magnitude rank from low to high values. The error bars correspond to the cumulative error arising from the two spectra used for quantification, obtained from signal-to-noise ratio estimates using NMRFAM-Sparky ($n = 1$). (C) Ubiquitination of TMEM106B was monitored using SDS-PAGE and immunoblotting as described above, comparing dimeric and monomeric variants. Note that although equal amounts of substrate were used, the antibody recognizes TMEM106B$^{C61S/C64S}$ more poorly, giving rise to weaker signals (compare SDS-PAGE and Western blots). Biophysical measurements were performed once, while biochemical experiments could be replicated at least three independent times in the laboratory. Source data are available online for this figure.

T cells to identify stable binding partners, and focused on the changes relative to the WT protein (Fig. 5B; Appendix Fig. S4). Consistent with the confocal microscopy data, we observed nearly no differences in the interactomes comparing the WT and K3A/K14A variants, but eight proteins were significantly less abundant in the C61S/C64S interactome (Fig. 5B, Dataset EV3). One of them was PDIA6, a protein disulfide isomerase, supporting the importance of reduced cysteines in allowing zinc coordination. Other proteins were mostly membrane-associated proteins, such as IFNGR1, TFRC, and HLA-A. We also identified the brain specific ITM2C (and ITM2B albeit with less confidence), which together with ITM2A constitute a subfamily of integral type II transmembrane glycoproteins of similar topology as TMEM106B, with links to the lysosomal pathway and neurodegenerative diseases (Martins et al, 2021; Namkoong et al, 2015). Altogether, our data show that mutation of the zinc coordination site impacts the biological function of TMEM106B both at the molecular and organelle levels, providing a possible link between the disruption of membrane protein–protein interactions and lysosomal regulation.

Neurons are particularly susceptible to deficiencies in the long-distance transport of vesicles and associated cargo, and many of the genes linked to neurodegeneration regulate the endolysosomal pathway (Grochowska et al, 2022; Morfini et al, 2009). We found TRIM2 to associate with endocytic components like SNF8, RABEP2, STAM, and HGS (Fig. 1, Dataset EV1). Among the proteins we identified through proximity labeling, we validated that the cytosolic domain of TMEM106B is ubiquitinated by TRIM2 and that the two proteins interact directly (Figs. 2 and 3). TMEM106B regulates the formation, acidification, and trafficking of lysosomes along microtubules in neurons and other cell types (Feng et al, 2022; Lüningschrör et al, 2020; Rademakers et al, 2021; Stroobants et al, 2021). At the molecular level, TMEM106B was shown to colocalize with the ESCRT-III component charged multivesicular body protein 2B (CHMP2B) (Jun et al, 2015), as well as microtubule associated MAP6 (also called STOP) (Schwenk et al, 2014). TRIM2 in turn can associate with microtubules and membrane phospholipids (Tsujita et al, 2010). Thus, there is evidence to support TRIM2 and TMEM106B existing at the interface of the microtubule cytoskeleton and the endolysosomal pathway through their functional or physical associations with the components that regulate vesicular trafficking. Although we could not establish a role for ubiquitination in regulating TMEM106B activity, we showed that at steady state, post-translational modifications at K3 and K14 do not dramatically affect the expression levels, localization, or function of TMEM106B as a lysosomal regulator, nor do they have a significant impact in protein–protein interactions. However, our conclusions are limited by the use of HEK 293-T cells, and future studies will be needed to assess the function of TMEM106B ubiquitination in neurons, where it is highly expressed and shown to have a strong phenotypic impact.

RING-type E3 ligases exert their biological roles through the formation of a quaternary complex that brings the E2 conjugating enzyme primed for ubiquitin transfer close to the target substrate. The ligase thus serves as an interaction platform that promotes catalysis (Yang et al, 2021). We show here that the interaction between TRIM2 and TMEM106B is direct and mediated by both the tripartite motif and the C-terminal domains, revealing a role for substrate recognition of the RBCC region of TRIM2 besides

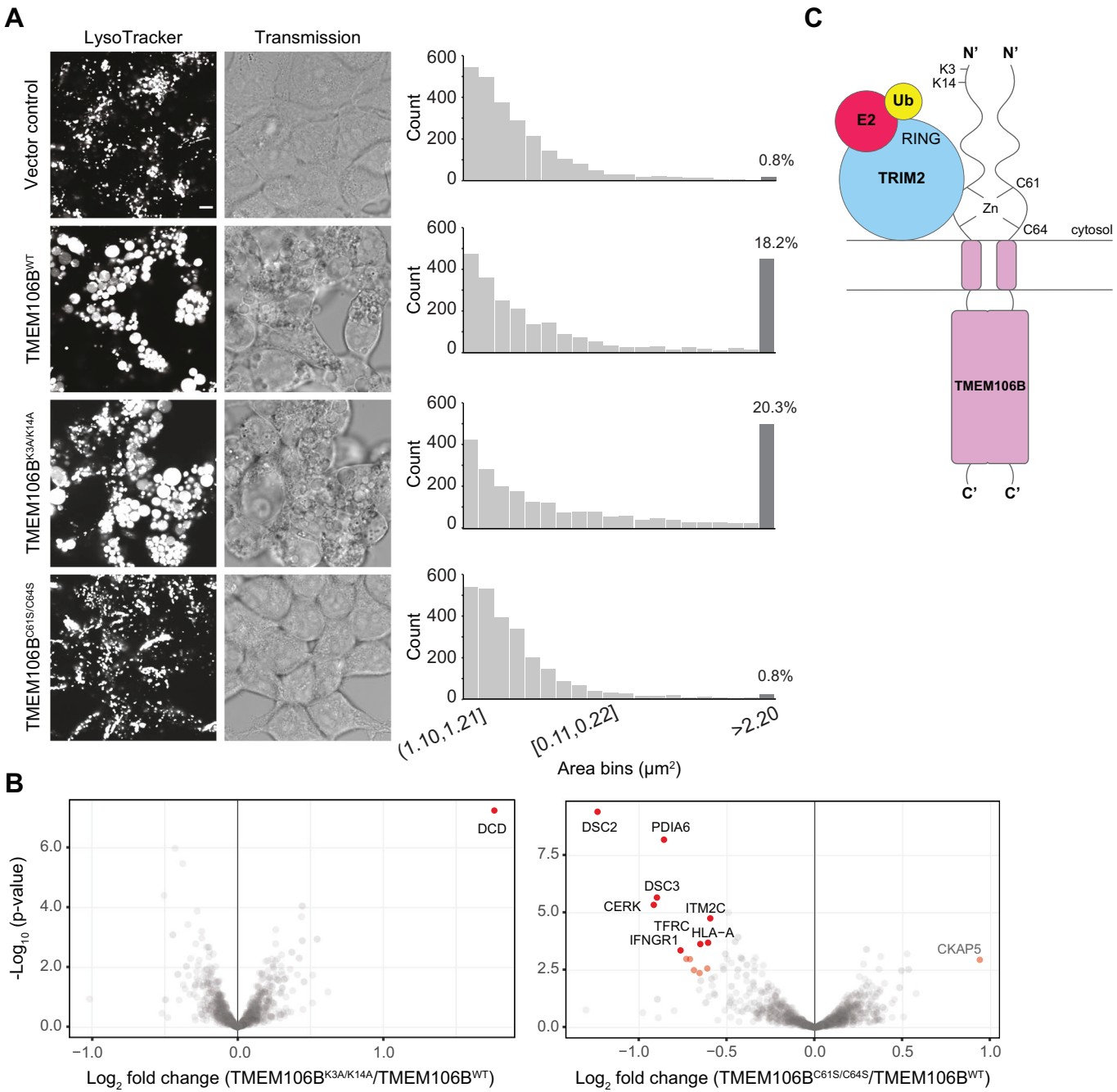

**Figure 5. Zinc binding is important in lysosomal size regulation by TMEM106B and establishing specific protein–protein interactions.**

(A) HEK 293-T cells were transfected with variants of full-length TMEM106B for ~48 h and incubated with LysoRed for ~45 min prior to live imaging. Two-dimensional images were collected in each case, and the lysosomal size distribution was quantified for ~2500 particles from >100 cells, selected in an unbiased manner, shown as a histogram on the right. Particles with an area above 2.2 μm² were binned together and are highlighted in darker gray, with the percentage of the total count shown above. The scale bar applies to all images shown and corresponds to 5 μm. (B) Volcano scatter plots comparing the protein interactome of TMEM106B variants relative to WT from co-immunoprecipitation and quantitative MS experiments from three biological replicates. The p-values were calculated using limma's empirical Bayes-moderated t-test. (C) Diagram of the current working model for the assembly of the complex leading to TMEM106B ubiquitination. The RING domain of TRIM2 associates with the ubiquitination machinery using a canonical interface. The remaining domains contribute to TMEM106B recruitment, which occurs via a zinc-mediated homodimerization surface in the cytosolic region involving cysteine residues 61 and 64. Lysine residues 3 and 14 do not adopt secondary structure and are ubiquitinated in vitro. Source data are available online for this figure.

catalysis (Fig. 3). While mostly unstructured, cytosolic TMEM106B contains a TRIM2-binding interface that is ordered around the metal-coordinating cysteine residues C61 and C64. The structure of the zinc-binding motif is characterized by two consecutive turns of the protein backbone that result in an S-shaped conformation (Fig. 3). The two TMEM106B monomers thus "sandwich" the metal ion, in a manner reminiscent of the zinc-hook found in Rad50 (Hopfner et al, 2002). Unlike Rad50, however, TMEM106B lacks secondary structural elements that could further stabilize the dimerization.

The dimerization of TMEM106B and its association with TRIM2 is disrupted upon mutation of the cysteine residues C61 and C64 (Fig. 4), and thus structural complementarity is important to the interaction between the E3 ligase and the substrate. Zinc binding and dimerization was also essential to the increase in lysosomal size caused by TMEM106B overexpression and to the formation of specific protein–protein interactions at the membrane (Fig. 5), which raises the possibility that the availability of this metal ion has a regulatory role through control of the monomer-dimer equilibrium, especially in the context of neurons where zinc concentrations are tightly regulated (Shannon et al, 2003). TMEM106B was previously shown to interact with paralogous TMEM106C (Stagi et al, 2014). We observed that the cytosolic regions of both TMEM106A and TMEM106C contain cysteine residues that could participate in metal coordination, and thus heterodimerization of TMEM106A/B/C paralogues may occur through the same mechanism as homodimerization. Previous studies have reported molecular weights of full-length cellular TMEM106B ranging from ~40 to 100 kDa, depending on the experimental protocol for sample preparation and the extent of glycosylation (Brady et al, 2013; Chen-Plotkin et al, 2012; Held et al, 2025; Takahashi et al, 2024). Heterodimerization may also impact the apparent molecular weight observed for TMEM106B.

While the details of complex assembly leading to TMEM106B ubiquitination remain to be fully elucidated, we present a working model that summarizes our data and provides key insight into how this can happen (Fig. 5C). The RING domain of TRIM2 engages with the ubiquitin machinery through an evolutionarily conserved interface to promote ubiquitination (Esposito et al, 2022). The disordered cytosolic N-terminus of TMEM106B (residues ~1–53) provides accessible lysine residues where ubiquitination can occur. The ordered zinc motif (~54–74) is responsible for the interaction with TRIM2, while the C-terminus of cytosolic TMEM106B (~75–95) does not adopt secondary structure in vitro. While the RING domain promotes ubiquitination, the remainder of TRIM2 recruits TMEM106B in a manner that is not restricted to one individual region but involves both N- and C-terminal domains. Not shown in the schematic is the fact that TRIM2 dimerization can take place both through the RING and the coiled-coil domains (Esposito et al, 2022). Thus, extensive oligomerization may occur as a function of TRIM2 concentration. Disordered linkers of ~20 and ~50 amino acids separate the RING and NHL domains from their neighbouring domains, providing additional flexibility. Various lysine residues present in the latter region can also be modified due to autoubiquitination (Dataset EV2). These features highlight the complexity of ubiquitin signaling complexes that may include variable states arising from PTMs and oligomerization, and future research will be essential to better define the details of this dynamic assembly.

# Methods

## Reagents and tools table

| Reagent/Resource | Reference or Source | Identifier or Catalog Number |
|---|---|---|
| **Experimental models** | | |
| HEK-293T cells (*H. sapiens*) | DSMZ - German Collection of Microorganisms and Cell Cultures | ACC: 635 |
| **Recombinant DNA** | | |
| pcDNA5-(-2)AP-Ubiquitin | (Mukhopadhyay et al, 2024) | |
| pEHA-C1-HA-BirA-TRIM2 WT | This study | |
| pEHA-C1-HA-BirA-TRIM2 C23S | This study | |
| pEHA-C1-HA-BirA-TRIM2 C60S | This study | |
| pEHA-C1-HA-TRIM2 WT | This study | |
| pEHA-C1-HA-TRIM2 C23S | This study | |
| pEHA-C1-HA-TRIM2 C60S | This study | |
| pCDNA3.1-flag-TMEM106B WT | This study | |
| pCDNA3.1-flag-TMEM106B K3A/K14A | This study | |
| pCDNA3.1-flag-TMEM106B C61S/C64S | This study | |
| pETM11-TRIM2 | This study | |
| pETM41-TRIM2[RBCC] | This study | |
| pETM22-TRIM2[FIL-NHL] | This study | |
| pETM11-Ubiquitin | This study | |
| pET28-MHL-UBE2D3 | This study | |
| pETM41-TMEM106B[1-95] | This study | |
| pET28-UBA1 | Addgene | Plasmid #32534 |
| **Antibodies** | | |
| Goat anti-mouse-HRP | Abcam | 97040 |
| Goat anti-rabbit-HRP | Abcam | 97051 |
| Goat anti-mouse-Alexa-488 | Abcam | 150113 |
| Anti-HA-tag | Abcam | 9110 |
| Anti-α-tubulin | Merck/Sigma-Aldrich | T6199 |
| Anti-TMEM106B | Biomol | A303-439A |
| Anti-flag-HRP | Sigma-Aldrich | A8592 |
| Anti-flag (monoclonal M2) | Sigma-Aldrich | F1804 |
| Anti-ubiquitin | Santa Cruz | P4D1 |
| Anti-AviTag | Genscript | A00674 |
| **Oligonucleotides and other sequence-based reagents** | | |
| PCR primers | This study | Table EV1 |
| **Chemicals, Enzymes and other reagents** | | |
| DMEM | ThermoFisher | Cat. #31885023 |

| Reagent/Resource | Reference or Source | Identifier or Catalog Number |
|---|---|---|
| FBS | ThermoFisher | Cat. #10270106 |
| PenStrep | ThermoFisher | Cat. #15140122 |
| L-glutamine | ThermoFisher | Cat. #A2916801 |
| Trypsin-EDTA | ThermoFisher | Cat. #25200056 |
| PR-619 (DUB inhibitor) | Merck/Sigma-Aldrich | Cat. #662141 |
| EndoFree MaxiPrep kit | Qiagen | Cat. #12362 |
| Biotin | Sigma-Aldrich | Cat. #B4501 |
| Fugene HD transfection reagent | Promega | Cat. #E2311 |
| Carfilzomib/PR-171 (Proteasome inhib.) | Selleckchem | Cat. #S2853 |
| N-Ethylmaleimide | Sigma-Aldrich | Cat. #E3876 |
| Complete EDTA-free protease inhibitor cocktail | Sigma-Aldrich | Cat. #11873580001 |
| Trypsin | Promega | Cat. #V5111 |
| TMT10plex isobaric label reagent | ThermoFisher | Cat. #90110 |
| LysoRed | Abcam | Cat. #112137 |
| FluoroBright™ DMEM | Gibco | Cat. #A1896701 |
| 4% paraformaldehyde | ThermoFisher | Cat. #28908 |
| Triton X-100 | Sigma-Aldrich | Cat. #T8787 |
| BSA | Sigma-Aldrich | Cat. #A7906 |
| glycine | Sigma-Aldrich | Cat. #50046 |
| ROTI®Mount FluorCare | Carl-Roth | Cat. #HP19.1 |
| E. coli BL21 (λDE3) | EMBL | N/A |
| ammonium bicarbonate | Sigma-Aldrich | Cat. #A6141 |
| chloroacetamide | Merck/Sigma-Aldrich | Cat. #C0267 |
| synthetic peptide TMEM106B[54-92] | ProteoGenix SAS | N/A |
| 3X FLAG® Peptide | Merck/Sigma-Aldrich | Cat. #F4799 |
| Anti-FLAG® M2 Magnetic beads | Sigma-Aldrich | M8823 |
| **Software** | | |
| IsobarQuant | (Franken et al, 2015) | |
| Mascot (v2.2.07) | (Perkins et al, 1999) | |
| R programming language | (R Core Team, 2019) | |
| Enrichr | (Kuleshov et al, 2016) | |
| Fiji/ImageJ | (Schindelin et al, 2012) | |
| FragPipe version 21.1 | (Kong et al, 2017) | |
| MSFragger 4.0 | (Kong et al, 2017) | |
| NMRpipe | (Delaglio et al, 1995) | |
| NMRFAM-Sparky | (Lee et al, 2015) | |
| CARA | http://cara.nmr.ch | |
| NMRView | (Johnson, 2004) | |

| Reagent/Resource | Reference or Source | Identifier or Catalog Number |
|---|---|---|
| CYANA 3.98.15 | (Güntert and Buchner, 2015) | |
| TALOS+ | (Shen et al, 2009) | |
| AlphaFold2-Multimer | (Evans et al, 2022) | |
| Astra 8.2.0 software | Wyatt Technology | |
| **Other** | | |
| TC20 cell counter | BioRad | N/A |
| Falcon® 60/100 mm TC-treated cell culture dishes | Corning | Cat. #353003 |
| T-25/T-25 cell culture flasks | ThermoFisher | Cat. #156472, 156340 |
| BioRuptor Plus sonicator | Diagenode | N/A |
| Pierce BCA protein assay kit | ThermoFisher | Cat. #23227 |
| Pierce Streptavidin magnetic beads | ThermoFisher | Cat. #88816 |
| SDS-PAGE Mini-Protean TGX gel | BioRad | Cat. #4561096 |
| Trans-Blot Turbo transfer system | BioRad | N/A |
| OASIS® HLB µElution Plate | Waters | Cat. #186001828BA |
| Agilent 1200 Infinity high-performance liquid chromatography system | Agilent | N/A |
| Gemini C18 column | Phenomenex | N/A |
| UltiMate 3000 RSLC nano LC system | Dionex | N/A |
| NanoEase™ M/Z HSS T3 column | Waters | N/A |
| Orbitrap Fusion™ Lumos™ Tribrid™ Mass Spectrometer | ThermoFisher | N/A |
| Pico-Tip Emitter | CoAnn Technologies | N/A |
| Glass-bottom dishes | Greiner Bio-One | Cat. #627871 |
| Leica SP8 confocal microscope system with a 63x/1.4 NA | Leica | N/A |
| Ni-NTA column (HisTrap HP) | Cytvia | Cat. #GE17-5247-01 |
| HiLoad 16/600 Superdex 75 or 200 columns | Cytvia | N/A |
| Glass vial for injection | VDS Optilab | Cat. #93908556 |
| Bruker Avance III 600 MHZ spectrometer | Bruker | N/A |
| Bruker Avance III 700 MHz spectrometer | Bruker | N/A |
| Bruker Avance III HD 1 GHz spectrometer | Bruker | N/A |
| NMR tubes | Norell | Cat. #ST500-7 |
| Amicon Ultra-15 3 kDa MWCO centrifugal filters | Sigma-Aldrich | Cat. #UFC9003 |

| Reagent/Resource | Reference or Source | Identifier or Catalog Number |
|---|---|---|
| Superdex 200 Increase 5/150 GL gel-filtration column | Cytiva | N/A |
| Agilent 1260 Infinity II HPLC system | Agilent | N/A |
| MALS detector MiniDAWN and Optilab | Wyatt Technology | N/A |

## HEK 293-T cell culture

HEK 293-T cells were obtained from the German Collection of Microorganisms and Cell Cultures (DSMZ, ACC: 635), tested for mycoplasma infection, and maintained in T-75 or T-25 cell culture flasks (Nunc EasYFlasks ThermoFisher, 156472, 156340) at 37 °C and 5% $CO_2$, using DMEM + 1 g/L D-glucose, L-glutamine, + pyruvate (ThermoFisher, 31885023) and supplemented with 10% heat-inactivated FBS, qualified from Brazil (ThermoFisher, 10270106), 1% PenStrep (ThermoFisher, 15140122), and 1% L-glutamine (ThermoFisher, A2916801). The cells were maintained and seeded according to the manufacturer's instructions using trypsin-EDTA (ThermoFisher, 25200056). Quantification of cell numbers and viability was achieved using a TC20 automated cell counter (Bio-Rad Laboratories GmbH, DE). Where indicated, deubiquitinase (DUB) inhibition consisted of a treatment for 2.5 h with a final concentration of 25 μM of PR-619 (Merck/Sigma-Aldrich, 662141).

## Mammalian expression plasmids

All plasmids used for mammalian transient transfections were prepared using endotoxin free (EndoFree) MaxiPrep kits (Qiagen no. 12362), and the primers were purchased from Sigma-Aldrich with standard desalting purification. The plasmids encoding HA-BirA-Rad18 and AviTag-Ubiquitin were kindly provided by Sagar Bhogaraju and have been described previously (Mukhopadhyay et al, 2024). The latter constitutes an optimized acceptor peptide with high selectivity named (−2)AP-Ub, in which two point mutations were introduced to change the sequence from GLNDI-FEAQKIEWHE to KGNDIFEAQKIEWHE (Mukhopadhyay et al, 2024). To generate HA-BirA-TRIM2 plasmids, restriction-free approaches were used (Bond and Naus, 2012). The catalytically dead variants contained the following point mutations introduced using site directed mutagenesis: C23S or C60S (Reikofski and Tao, 1992). We found no significant differences in the expression levels of these mutants and thus used the C60S variant in the proximity labeling approach as the catalytically inactive variants of TRIM2 ("dead"). Plasmids lacking the BirA gene, including HA-TRIM2$^{WT}$, HA-TRIM2$^{C60S}$, and HA-TRIM2$^{C23S/C60S}$ were derived from those above and generated using standard cloning approaches to amplify the regions of interest. The gene for TMEM106B was obtained through Addgene (plasmid #179385, UniProt ID: Q9NUM4). Cloning of 3x flag-tagged TMEM106B$^{WT}$, TMEM106B$^{K3A/K14A}$ and TMEM106B$^{C61S/C64S}$ into pCDNA3.1 was accomplished using similar methods.

## Transient co-transfections, lysis, and pull-down in substrate identification

Three million HEK 293-T cells were seeded in Falcon® 100 mm TC-treated cell culture dishes (Corning, 353003) and maintained as described above. The following day (~16 h), each dish was supplemented with a final concentration of 100 μM biotin (Sigma-Aldrich, B4501) and shortly thereafter co-transfected with 5 μg of each of the BirA-containing and AviTag-Ub plasmids, using Fugene HD transfection reagent (Promega, E2311) diluted in DMEM according to manufacturer's instructions. Twenty-four hours post-transfection, the proteasome was inhibited with a final concentration of 0.5 μM Carfilzomib/PR-171 (Selleckchem, S2853) for 5 h. The cells were collected 29 h post-transfection using plastic cell scrapers after briefly washing with ice-cold 1x PBS. The resulting cell suspension was centrifuged for 5 min at 3000 r.c.f. and 4 °C, and the pellet frozen at −80 °C until further processing.

To lyse, the cell pellet was thawed on ice for 15 min and resuspended in 500 μL of ice-cold lysis buffer (50 mM Tris-HCl, pH 7.4, 0.1% SDS, 1% Triton X-100, 500 mM NaCl, 1 mM EDTA, 1.5 mM $MgCl_2$, 0.5 mM TCEP, 10 mM N-Ethylmaleimide (NEM, Sigma-Aldrich, E3876), 1 μg/mL RNAse A (Qiagen), 1 μg/mL DNAse I (Roche), and 1% Complete EDTA-free protease inhibitor cocktail tablet (Sigma-Aldrich, 11873580001), followed by a 20 min incubation at 37 °C to allow DNA and RNA digestion. The cell suspension was then cooled on ice for 10 min and lysed using a BioRuptor Plus sonicator (Diagenode) equilibrated at 4 °C, using 6 cycles of 30 s ON, 30 s OFF, with low power setting. The resulting lysate was centrifuged at 13,000 r.c.f. for 10 min at 4 °C, and the supernatant used for further analysis. The total protein concentration of each sample (soluble fraction) was determined using the Pierce BCA protein assay kit according to manufacturer's instructions (ThermoFisher, 23227) and normalized to the lowest protein concentration using lysis buffer. Twenty microliters of Pierce Streptavidin magnetic beads (ThermoFisher, 88816) were incubated with 450 μL of each normalized cleared lysate and rotated at 4 °C for 2 h to allow enrichment of biotinylated ubiquitin moieties. The beads were washed three times for 20 min each, with rotation at 4 °C, using 750 μL of wash buffer (50 mM Tris-HCl, pH 7.42, 0.1% SDS, 1% Triton X-100, 500 mM NaCl, 1 mM EDTA, 1.5 mM $MgCl_2$, 0.5 mM TCEP, 10 mM NEM, 10 mM of freshly prepared dithiothreitol (DTT), and 1% Complete EDTA-free protease inhibitor cocktail tablet). The bound proteins were eluted using 100 μL of elution buffer consisting of 100 mM Tris-HCl, pH 6.5, 2% SDS, 200 mM DTT, 3 mM bromophenol blue, 2.15 M glycerol, 25 mM biotin, and boiled for 5 min at 95 °C. Two biological and two technical replicates were conducted in each case (n = 4). The biological and technical replicates did not show differences in variability and were therefore treated equally during data analysis.

## Immunoblotting

The cleared lysate from cellular experiments or samples from in vitro assays (~2–50 μg) were run on a gradient denaturing SDS-PAGE gel (Mini-Protean TGX, BioRad) and transferred to a polyvinylidene difluoride membrane using the Trans-Blot Turbo transfer system according to manufacturer's instructions (BioRad).

The membrane was treated with a blocking buffer containing 5% milk powder and 0.1% Tween (Sigma-Aldrich) dissolved in 1x PBS (10 mM NaH$_2$PO$_4$, pH 7.4, 2.7 mM KCl, 1.5 mM KH$_2$PO$_4$, and 137 mM NaCl) and incubated at room temperature for 30 min while shaking. The membrane was then incubated with the primary antibodies listed below, diluted in blocking buffer for ~16 h at 4 °C while mixing. The membranes were washed with 0.1% Tween/1x PBS three times for 5 min each at room temperature, and the secondary antibodies (goat anti-mouse-HRP, Abcam 97040 and goat anti-rabbit-HRP, Abcam 97051) diluted in blocking buffer 1:5000 and incubated for ~1 h at room temperature. After similar washes, the membranes were treated with horseradish peroxidase substrate according to manufacturer's instructions (Merck-Milli-pore Immobilon Western), and the chemiluminescence signal visualized using a BioRad ChemiDoc system. The following antibodies and dilutions were used in each case: anti-HA-tag antibody (Abcam 9110) 1:4000, anti-α-tubulin (Merck/Sigma-Aldrich, T6199) 1:5000, anti-TMEM106B (Biomol, A303-439A) 1:2000–1:5000, anti-flag (Sigma-Aldrich, A8592) 1:5000, anti-ubiquitin (Santa Cruz, P4D1) 1:2000–1:4000, and anti-AviTag (Genscript, A00674) 1:3000.

## Quantitative mass spectrometry in substrate identification

Following elution from the streptavidin beads, the samples were further processed by reducing the disulfide bridges of cysteine-containing proteins with DTT (10 mM dissolved in 50 mM HEPES, pH 8.5 for 30 min at 56 °C). The reduced cysteines were alkylated with 2-chloroacetamide (20 mM in 50 mM HEPES, pH 8.5) at room temperature and in the dark for 30 min. The samples were then submitted to the SP3 protocol (Hughes et al, 2019) and trypsin was added in a ratio of 1 to 50 (enzyme to protein) for overnight digestion at 37 °C (sequencing grade, Promega, V5111). The following day, the peptides were recovered using 50 mM HEPES (pH 8.5) by collecting the supernatant from the beads and combining it with a new wash with the same buffer. Peptides were labeled with the TMT10plex isobaric label reagent (ThermoFisher, 90110) according to the manufacturer's instructions (Werner et al, 2014). Subsequently, the samples were combined for multiplexing and additional clean-up steps using an OASIS® HLB µElution Plate (Waters, 186001828BA). Offline high pH reverse phase fractionation was carried out on an Agilent 1200 Infinity high-performance liquid chromatography system, equipped with a Gemini C18 column (Reichel et al, 2016) (3 µm, 110 Å, 100 × 1.0 mm, from Phenomenex).

LC-MS/MS was carried out using an UltiMate 3000 RSLC nano LC system (Dionex) fitted with a trapping cartridge (µ-Precolumn C18 PepMap 100, 5 µm, 300 µm i.d. × 5 mm, 100 Å) and an analytical column (nanoEase™ M/Z HSS T3 column 75 µm × 250 mm C18, 1.8 µm, 100 Å, Waters). Trapping was carried out with a constant flow of trapping solution (0.05% trifluoroacetic acid in water) at 30 µL/min into the trapping column for 6 min. Subsequently, peptides were eluted while running solvent A (0.1% formic acid in water, 3% DMSO) with a constant flow of 0.3 µL/min, with increasing percentage of solvent B (0.1% formic acid in acetonitrile, 3% DMSO). The outlet of the analytical column was coupled directly to an Orbitrap Fusion™ Lumos™ Tribrid™ Mass Spectrometer (ThermoFisher) using the Nanospray Flex™ ion source in positive ion mode.

The peptides were introduced into the instrument via a Pico-Tip Emitter (360 µm OD × 20 µm ID; 10 µm tip, CoAnn Technologies) and an applied spray voltage of 2.4 kV. The capillary temperature was set to 275 °C. Full mass scans were acquired in the range of 375–1500 $m/z$ in profile mode with a resolution of 120,000. The filling time was set to a maximum of 50 ms with a limitation of $4 \times 10^5$ ions. Data-dependent acquisition was performed with the resolution set to 30,000, with a fill time of 94 ms and a limitation of $1 \times 10^5$ ions. A normalized collision energy of 38 was applied. MS$^2$ data was acquired in profile mode.

## Mass spectrometry data analysis in substrate identification

IsobarQuant and Mascot (v2.2.07) were used to process the acquired data, which was searched against a Uniprot Homo sapiens (UP000005640) proteome database containing common contaminants and reversed sequences (Franken et al, 2015). The following modifications were included in the search parameters: carbamido-methyl (C) and TMT10 (K) as fixed modifications; as well as acetyl (N-term), oxidation (M), and TMT10 (N-term) as variable modifications. The following error tolerances were set: 10 ppm for MS$^1$ scans and 0.02 Da in the case of MS$^2$. In addition, trypsin was set as the protease with a maximum of two missed cleavages allowed, the minimum peptide length was set to seven amino acids, and at least two unique peptides were required for protein identification. The false discovery rate (fdr) at the peptide and protein level was set to 1%.

The raw output files of IsobarQuant (protein.txt) were processed using the R programming language (R Core Team, 2019). Only the proteins which were identified in two out of two technical replicate runs were kept. Raw TMT intensities ('signal_sum' columns) were first cleaned for batch effects using limma (Ritchie et al, 2015) and further normalized using variance stabilization normalization (Huber et al, 2002). Missing values were imputed with the knn method using the Msnbase package (Gatto and Lilley, 2012). Proteins were tested for differential expression using the limma package. The replicate information was added as a factor in the design matrix given as an argument to the 'lmFit' function of limma. In addition, imputed values were given a weight of 0.05 in the 'lmFit' function. A protein was annotated as a hit when the fdr was less than 5% and the fold-change was at least 100%, while protein candidates had an fdr less than 5% and a fold-change of at least 40%.

Proteins of interest (POI = TRUE, Dataset EV1) are highlighted in red in Fig. 1 and fulfilled the following criteria: (i) an enrichment in the WT condition relative to catalytically dead of at least 40% with a fdr of 5% or less, (ii) identical enrichment and fdr in the WT versus BirA-only comparison, and (iii) they were not significantly different in the catalytically dead versus BirA-only conditions. This resulted in a set of 29 candidate proteins that were further considered for analysis.

## TMEM106B co-immunoprecipitations and interactome analyses

A similar procedure as described above for the transfection and lysis of TRIM2-expressing cells was followed, and we describe briefly the changes. Cells at ~50% confluency were transfected with

5 µg of TMEM106B plasmid DNA using the Fugene HD transfection reagent diluted in Opti-MEM. Twenty-eight hours post-transfection, the cells were collected and frozen at −80 °C prior to downstream analysis. The lysis buffer contained 150 mM NaCl, no EDTA or $MgCl_2$, and benzonase (EMBL) rather than DNAse/RNAse. All other components remained the same. Following lysis and normalization of protein concentrations, 60 µL of magnetic flag beads (Sigma-Aldrich, M8823) were incubated with ~350 µg of protein while rotating for 16 h, at 4 °C. The beads were washed at 4 °C three times for 5–7 min each, with 200 µL of TBS (50 mM Tris, 150 mM NaCl, pH 7.5) using a magnetic sample holder. The interacting proteins were eluted with 0.2 mg/mL of 3X FLAG® Peptide (Sigma-Aldrich, F4799) diluted in TBS. Three biological replicates were collected for each sample type: WT, K3A/K14A, C61S/C64S, and a negative control. Downstream processing for quantitative MS and enrichment analysis was performed as above. The results are shown in Fig. 5B, Dataset EV3, and Appendix Fig. S4. Proteins were considered a "hit" with an fdr <0.05 and a fold-change of at least 1.5-fold; and a 'candidate' with an fdr <0.2 and a fold change of at least 1.5-fold.

### Enrichment analysis

The gene symbols of protein candidates were used as input for the webserver program Enrichr (Kuleshov et al, 2016). In cases of ambiguity in the protein identity (i.e., HSPA1B|HSPA1A), only the first annotation was retained.

### Live confocal imaging and immunofluorescence

HEK 293-T cells cultured in glass-bottom dishes (Greiner Bio-One) were transfected one day after passaging with 0.5–2 µg of plasmid DNA using Fugene HD (Promega) according to manufacturer's instructions. The lysosomal marker LysoRed (Abcam #112137) was used according to manufacturer's instructions and incubated for ~45 min 48 h post transfection. Rather than washing with HHBS, the cells were incubated with FluoroBright™ DMEM (Gibco, A1896701). To quantify the size distribution of lysosomes, 2D images collected at a resolution of $2048 \times 2048$ pixels ($184 \times 184$ $\mu m^2$) were processed in Fiji/ImageJ (Schindelin et al, 2012). The Otsu threshold was applied automatically, and the images converted to binary with the watershed transformation to separate lysosomes in proximity. The area of each of the resulting ~2500 particles was analyzed, excluding those smaller than 0.1 $\mu m^2$, larger than 100 $\mu m^2$, as well as those with a circularity lower than 0.8. An equal number of particles, selected in an unbiased manner from five images containing in total >100 cells were quantified and plotted as a histogram for each condition. Particles with an area above 2.2 $\mu m^2$ were binned together to highlight the increase in size resulting from TMEM106B overexpression. These experiments were replicated a minimum of 3 times in the laboratory with the same pattern of results, with higher DNA concentrations and longer transfection times resulting in more accentuated phenotypes.

To perform fluorescent immunostaining of flag-tagged TMEM106B variants, similar procedures for culturing and transfection were applied. The following steps occurred at room temperature using 1x PBS as the buffer solution unless otherwise noted. The cells were fixed for 10 min with 4% paraformaldehyde (ThermoFisher, 28908). The fixative was washed 3 times, and the cells permeabilized for 15 min using 0.25% Triton X-100 (Sigma-Aldrich, T8787). After similar washes, the samples were blocked with 1% BSA (Sigma-Aldrich, A7906) and 22.5 mg/mL glycine (Sigma-Aldrich, 50046) for 30 min. The cells were washed once, and incubated with the primary anti-flag antibody (Sigma-Aldrich, F1804) diluted 1:1000 in 1% BSA over ~16 h at 4 °C. The following day, the samples were washed three times, and the secondary antibody (Alexa-488 conjugate, Abcam, 150113) was incubated for 1 h, diluted 1:1000 in 1% BSA and protected from light. The cells were counterstained with a mounting solution containing DAPI (ROTI®Mount FluorCare from Carl-Roth). In all cases, the samples were imaged using a Leica SP8 confocal microscope system with a 63x/1.4 numerical aperture oil objective using lasers at wavelengths of 405, 488, 552, and/or 638 nm.

### Recombinant protein expression and purification

The genes encoding human TRIM2 and ubiquitin (NCBI gene IDs: Q9C040, P0CG48) were cloned into the expression plasmid pETM11 (generated at EMBL); while the gene encoding UBE2D3 (ID: P61077) was cloned into pET28-MHL (Addgene, plasmid #26096). Human TMEM106B[1-95] and its variants (ID: Q9NUM4), as well as TRIM2[RBCC] (residues 8–318) were cloned into pETM41 (EMBL), while TRIM2[FIL-NHL] (residues 318–744) was cloned into pETM22 (EMBL). The vectors produced encoded poly-histidine ($his_6$) affinity tags used for purification with Ni-NTA Sepharose columns that could be removed by proteolytic cleavage with TEV or 3 C proteases, except in the case of UBA1. pETM41 and pETM22 contained additional maltose-binding protein and thioredoxin solubility tags, respectively. Cloning involved standard restriction free methods with the primers listed in Table EV1 (Bond and Naus, 2012). All plasmids were transformed into strain BL21 (λDE3) of E. coli for recombinant expression. Protein production was induced with 0.2 mM isopropyl β-d-1-thiogalactopyranoside (IPTG) over ~16 h at 20 °C, at optical densities (600 nm) of 0.6 to 1.5. The cells were grown in Lysogeny broth (LB) or Terrific broth (TB), while isotopically labeled proteins used for NMR spectroscopy were produced in M9 minimal media supplemented with 2 g/L $^{13}C_6$-D-glucose and/or 0.5 g/L $^{15}NH_4Cl$ as sole carbon and nitrogen sources, respectively. Following expression, the cells were centrifuged at 3000 r.c.f. at 4 °C, and the pellet frozen at −20 °C until further processing. The cells were then thawed and resuspended in lysis buffer (20 mM $Na_2HPO_4$, 500 mM NaCl, 40 mM imidazole, pH 7.4), supplemented with 1 mg/mL lysozyme, 25 ng of benzonase, 0.5 mM TCEP, and 1x cOmplete™ Mini EDTA-free Protease Inhibitor Cocktail (Sigma-Aldrich), followed by incubation at room temperature for ~30 min. The cells were lysed using a homogenizer equilibrated at 4 °C, and the cleared supernatant applied to a Ni-NTA column (HisTrap HP, Cytvia) to allow binding. After washing with ~20 column volumes, the his-tagged proteins were eluted with 20 mM $Na_2HPO_4$ (pH 7.4), 500–1000 mM NaCl, and 250–500 mM imidazole. The appropriate fractions were pooled and dialyzed against 50 mM Tris (pH 7.5), 250 mM NaCl, and 1 mM DTT over ~16 h at 4 °C, with addition of 1–2 mg of his-tagged TEV or 3 C proteases. A second Ni-NTA chromatography step was performed to separate the cleaved protein, and a final size-exclusion chromatography (SEC) step was included in all cases using HiLoad 16/600 Superdex 75 or 200 columns (Cytvia). In the case of proteins prepared for

ubiquitination assays, the SEC buffer consisted of 50 mM HEPES (pH 7.5), and 150 mM NaCl. TRIM2 and TMEM106B preparations included 0.5 mM DTT and 10% glycerol additionally. For NMR spectroscopic studies, the SEC buffer consisted of 20 mM sodium phosphate (pH 6.5), 100 mM NaCl, and 0.5 mM TCEP.

## In vitro ubiquitination assays

Protein stocks obtained as described above were diluted with reaction buffer (50 mM HEPES, 150 mM NaCl, pH 7.5) to the following final concentration ranges: 150–450 µM ubiquitin, 50–70 µM E2, 0.8–1.3 µM E1, 0.7–1 µM TRIM2, and 5–80 µM TMEM106B$^{1-95}$. The reaction was started with the addition of a final concentration of 5 mM ATP-MgCl$_2$, followed by a 30-minute incubation at 37 °C. The reaction was quenched subsequently with a final concentration of 50 mM EDTA diluted in reaction buffer, and the resulting mixture analyzed by denaturing SDS-PAGE or immunoblotting. The E2 chosen (UBE2D3) is closely related to UBE2D1 and UBE2D2. Members of this family of E2s are known to function with RING ligases, and specifically with TRIM2 (Esposito et al, 2022). TRIM2 autoubiquitination was observed with all three E2s.

## Identification of ubiquitination sites

The assayed samples derived from in vitro experiments were applied to SDS-PAGE gels and visualized by Coomassie blue staining. The corresponding regions of interest were excised from the gel, cut into small pieces (~1 mm × 1 mm), and transferred to 0.5 ml Eppendorf tubes. In all following steps, each buffer was exchanged by two consecutive 15-minute incubations of the gel pieces with 200 µL of acetonitrile (ACN), followed by ACN removal. Proteins were reduced by the addition of 200 µL of aqueous solution consisting of 10 mM DTT and 100 mM ammonium bicarbonate (AmBiC, Sigma-Aldrich, A6141), then incubated at 56 °C for 30 min. Proteins were alkylated by the addition of 200 µL of aqueous solution of 55 mM chloroacetamide (CAA, Merck/Sigma-Aldrich, C0267), 100 mM AmBiC, and incubated for 20 min in the dark. A 0.1 µg/µL stock solution of trypsin (Promega, V511A) in trypsin resuspension buffer (Promega, V542A) was diluted with ice-cold 50 mM AmBiC aqueous solution to achieve a final concentration of 1 ng/µL. 50 µL thereof were added to gel pieces, which were incubated first for 30 min on ice and then overnight at 37 °C. Gel pieces were sonicated for 15 min, centrifuged briefly to collect them, and the supernatant was transferred into a glass vial for injection (VDS Optilab, 93908556). The remaining gel pieces were washed with 50 µL of an aqueous solution of 50% ACN and 1% formic acid and sonicated for 15 min. The combined supernatants were dried in a vacuum concentrator and reconstituted in 10 µL of an aqueous solution of 0.1% (v/v) formic acid.

Peptides were analyzed by LC-MS/MS using a similar strategy as described above. The following gradient program was used: from 4 to 8% solvent B in 6 min, 8 to 23% for a further 41 min, 23 to 38% in 5 min, followed by an increase of B to 80% for 4 min, and a re-equilibration to 2% solvent B for 4 min. The Orbitrap Fusion Lumos was operated in positive ion mode with a spray voltage of 2.4 kV and capillary temperature of 275 °C. Full scan MS spectra with a mass range of 375–1200 $m/z$ were acquired in profile mode

using a resolution of 120,000, a maximum injection time of 50 ms, and an AGC target of $2 \times 10^5$. Precursors were isolated using the quadrupole with a window of 1.2 $m/z$. For fragmentation, HCD was used with a fixed collision energy of 34%. MS$^2$ spectra were acquired using the Orbitrap with a resolution of 15,000. The maximum injection time was set to 54 ms and the AGC target to $2 \times 10^6$.

The data acquired were analyzed using FragPipe version 21.1 and MSFragger 4.0 (Kong et al, 2017) using the E. coli protein sequence database (UniProt: UP000000625, 4402 entries, February 2022) that includes common contaminants, along with the sequences for human TRIM2 and TMEM106B. The following modifications were considered: carbamidomethyl (protein C-terminus, fixed), acetyl (protein N-terminus, variable), oxidation (methionine, variable), as well as Gly-Gly (lysine, variable). The mass error tolerance was set to 20 ppm for MS$^1$ as well as MS$^2$ spectra. A maximum of 3 missed cleavages were allowed. The minimum peptide length was seven amino acids. A false discovery rate below 0.01 was applied at the peptide and protein level.

## NMR spectroscopy

NMR experiments were collected on Bruker Avance III 600, 700 MHz, and Bruker Avance III HD 1 GHz spectrometers equipped with a room temperature probe (700 MHz) or cryoprobes (600 and 1000 MHz). The experiments were collected at 298 K (25 °C) with samples containing 10% D$_2$O as a locking agent. The NMR sample buffer consisted of 20 mM sodium phosphate (pH 6.5), 100 mM NaCl, and 0.5 mM TCEP, supplemented with Zn$^{+2}$ (i.e., 100–800 µM of ZnCl$_2$, or ZnSO$_4$ in case of samples derived from chemical synthesis). Isotopically $^{15}$N/$^{13}$C-labeled samples of TMEM106B$^{1-95}$ were obtained as described above from recombinant expression in E. coli and concentrated to 100–600 µM using Amicon Ultra-15 3 kDa MWCO centrifugal filters (Sigma-Aldrich), calculated according to the predicted extinction coefficient of TMEM106B$^{1-95}$ at 280 nm. An unlabeled synthetic peptide spanning residues 54–92 (TMEM106B$^{54-92}$) was purchased from ProteoGenix SAS (Schiltigheim, France) in lyophilized form at 95% purity, and resuspended in NMR sample buffer to a concentration of 0.6 mM supplemented with fivefold excess of Zn$^{+2}$. Chemical shift assignments were obtained using either standard $^1$H-$^{13}$C-$^{15}$N correlation experiments (Sattler et al, 1999) with apodization weighted sampling to reduce data collection times (Simon and Köstler, 2019) in the case of isotopically labeled TMEM106B$^{1-95}$, or homonuclear 2D $^1$H,$^1$H-COSY, TOCSY and NOESY, along with $^1$H-$^{13}$C/$^{15}$N-HSQC spectra at natural abundance in the case of TMEM106B$^{54-92}$. These have been deposited under BMRB accession codes 52589 and 34953, respectively. The raw data were processed and analyzed using NMRpipe (Delaglio et al, 1995), NMRFAM-Sparky (Lee et al, 2015), CARA (http://cara.nmr.ch), and NMRView (Johnson, 2004).

To test binding, samples containing equal quantities of $^{15}$N-labeled TMEM106B$^{1-95}$ were diluted either with buffer or TRIM2 variants prepared in the same conditions. The RBCC (residues 8–318) or FIL-NHL (318–744) regions of TRIM2 were added in a 5-fold molar excess, while full-length TRIM2 was added in a 1:1 ratio. Line broadening observed in $^1$H-$^{15}$N HSQC experiments resulting from binding was quantified from the height of well-resolved amide peaks, comparing the two types of samples.

The errors were estimated from the signal-to-noise ratio in each spectrum. In the case of TMEM106B[C61S/C64S], we did not assign the amide chemical shifts and the comparison with the WT variant was performed for all well-resolved peaks that could be quantified in both sample types. To test EDTA-mediated denaturation, [15]N-labeled TMEM106B[1-95] was treated with a final concentration of 5 mM EDTA and heated to 45 °C for 2 h, before lowering the temperature to 25 °C for data collection. Zinc binding was assessed with a sample containing a mixture of monomeric and dimeric species, before and after supplementation with equimolar (100 μM) amounts of $ZnCl_2$. The addition of a final concentration of 10 mM DTT was used to observe changes in TMEM106B conformation under reducing conditions.

## Structural ensemble calculations with CYANA

Both TMEM106B[1-95] and TMEM106B[54-92] gave rise to similar chemical shifts corresponding to the structured region (amino acids ~57–71). The remaining residues show random coil chemical shifts and only intra-residual and sequential NOEs. To reduce signal overlap resulting from the disordered regions, we thus focused on the shorter peptide. Distance restraints were obtained from 2D $^1H,^1H$-NOESY spectra (mixing time 150 ms). The NOE cross-peaks were assigned automatically during structure calculation using the noeassign function of CYANA 3.98.15, resulting in 145 intermolecular distance restraints defining the dimer interface (Güntert and Buchner, 2015). The assignments were manually validated. Symmetry restraints were applied to the Cα atoms of all residues in each of the two monomeric chains. The tetrahedral conformation of the Zn-($S^{Cys}$)$_4$ cluster was maintained using Zn-S and S-S upper and lower distance restraints of 2.25–2.35 Å and 3.75–3.85 Å, respectively. Dihedral angles were obtained from TALOS+ (Shen et al, 2009). All restraints were used to calculate a structure ensemble using CYANA. The resulting 20 models with lowest energy (i.e., using the target function) have been deposited under PDB accession code 9GI8. While residues 57 to 71 superimpose very well with a main chain RMSD of 0.1 Å, the remainder of the peptide is unstructured. See also Appendix Table S1 for structural statistics.

## Size exclusion chromatography–multi-angle light scattering (SEC-MALS)

TMEM106B[1-95] WT and C61S/C64S were prepared as described above for NMR experiments. A volume of 50 μL was injected onto a Superdex 200 Increase 5/150 GL gel-filtration column (Cytiva) on an Agilent 1260 Infinity II HPLC system (Agilent), at protein concentrations of 1 mg/mL (~0.3 mg/mL at the apex of the peak). The run was performed at room temperature and with a flow rate of 0.3 mL/min. The column was coupled to a MALS detector (MiniDAWN and Optilab, Wyatt Technology). Data were analyzed using the Astra 8.2.0 software (Wyatt Technology). The standard protein refractive index increment (dn/dc) value of 0.1850 was applied for molar mass determination. To determine metal-binding properties, a final concentration of 5 mM EDTA was added, and the sample heated to 45 °C for 2 h and cooled to room temperature before measurement. The negative controls consisted of samples that were heated in the absence of EDTA in the same manner or treated with EDTA only.

## AlphaFold2-Multimer

The amino acid sequence of human TMEM106B (residues 1–95) was used as input to predict a homodimer. The predictions of 25 structural models were performed with the AlphaFold2 (release version 2.3.2) multimer pipeline (model_preset=multimer) using the multimer_v3 parameter set and the UniRef30 database version 2023_02 (preprint: Evans et al, 2022). The maximum template release date was set to the future. Default values were used for db_preset (full_dbs) and num_recycle (20), and all models were selected for Amber relaxation. Shown in Fig. 3D are the top 5 ranked relaxed models after main chain alignment.

In all cases, no blinding of samples was done, no samples meeting quality standards were excluded from the analysis, and the EMBO Press reporting standards adapted from MDAR were followed.

## Data availability

The NMR-derived structure of TMEM106B[54-92] from this publication has been deposited to the PDB and assigned the identifier 9GI8 (https://www.rcsb.org/structure/9GI8). The chemical shift assignments of TMEM106B[1-95] and TMEM106B[54-92] can be found at the BMRB (https://bmrb.io) with the identifiers 52589 and 34953, respectively (https://bmrb.io/data_library/summary/index.php?bmrbId=52589 and https://bmrb.io/data_library/summary/index.php?bmrbId=34953). The mass spectrometry proteomics data have been deposited to the ProteomeXchange Consortium via the PRIDE (Perez-Riverol et al, 2022) partner repository with the dataset identifier PXD055297 (https://www.ebi.ac.uk/pride/archive/projects/PXD055297). New materials created in this study are available upon request.

The source data of this paper are collected in the following database record: biostudies:S-SCDT-10_1038-S44319-025-00667-3.

## Peer review information

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

## Acknowledgements

We are grateful to the Proteomics, Advanced Light Microscopy, and Protein Expression and Purification core facilities at EMBL, as well as HPC resources

through EMBL IT services for access to scientific instrumentation, plasmids, reagents, infrastructure, and training. We thank Sandra Augsten and Bernd Simon for advice on cell culture experiments and NMR spectroscopy, respectively; as well as Stefan Terjung and Jean-Karim Hériché for help with confocal microscopy. We thank members of the Hennig and Mahamid groups for fruitful discussions. C.P.-B. was supported by the EMBL Interdisciplinary Postdoc (EI$_3$POD) Program fellowship under Marie Sklodowska-Curie Actions COFUND (grant no. 664726). This study was supported by funding from the Deutsche Forschungsgemeinschaft (DFG) via an Emmy-Noether Fellowship (project number: 267437786) to J.H..

## Author contributions

**Cecilia Perez-Borrajero**: Conceptualization; Data curation; Formal analysis; Validation; Investigation; Visualization; Methodology; Writing—original draft; Project administration; Writing—review and editing. **Frank Stein**: Formal analysis; Visualization; Methodology; Writing—review and editing. **Kristian Schweimer**: Formal analysis; Investigation; Methodology; Writing—review and editing. **Mandy Rettel**: Formal analysis; Investigation; Methodology; Writing—review and editing. **Jennifer J Schwarz**: Formal analysis; Investigation; Methodology. **Per Haberkant**: Formal analysis; Investigation; Methodology. **Karine Lapouge**: Formal analysis; Investigation; Methodology. **Jesse Gayk**: Investigation. **Thomas Hoffmann**: Investigation; Methodology. **Sagar Bhogaraju**: Resources; Methodology. **Kyung-Min Noh**: Conceptualization; Supervision. **Mikhail Savitski**: Supervision; Methodology; Project administration. **Julia Mahamid**: Resources; Supervision; Project administration. **Janosch Hennig**: Conceptualization; Resources; Formal analysis; Supervision; Funding acquisition; Validation; Investigation; Visualization; Methodology; Project administration; Writing—review and editing.

Source data underlying figure panels in this paper may have individual authorship assigned. Where available, figure panel/source data authorship is listed in the following database record: biostudies:S-SCDT-10_1038-S44319-025-00667-3.

## Funding

## Disclosure and competing interests statement

The authors declare no competing interests.

