## [Peer Review File · EMBO Reports]

TRIM2 E3 ligase substrate discovery reveals zinc regulation of TMEM106B in the endolysosomal pathway

Cecilia Perez-Borrajero, Frank Stein, Kristian Schweimer, Mandy Rettel, Jennifer Schwarz, Per Haberkant, Karine Lapouge, Jesse Gayk, Thomas Hoffmann, Sagar Bhogaraju, Kyung Min Noh, Mikhail Savitski, Julia Mahamid, and Janosch Hennig

Corresponding author(s): Janosch Hennig (janosch.hennig@uni-bayreuth.de) , Cecilia Perez-Borrajero (cecilia.perez@embl.de)

Review Timeline:

Transfer Date:	10th Jun 25
Editorial Decision:	23rd Jun 25
Revision Received:	10th Sep 25
Editorial Decision:	7th Nov 25
Revision Received:	17th Nov 25
Accepted:	24th Nov 25

**Transaction Report: This manuscript was transferred to
EMBO reports following peer review at Review Commons.**

**Review
COMMONS**

Review #1

1. Evidence, reproducibility and clarity:

Evidence, reproducibility and clarity (Required)

I summarize the key strengths of the study and outline suggestions for key analysis / experiment that would strengthen the study below.

Ubiquitin E3 ligases provide specificity to the Ub system; however for most E3s substrates are poorly characterized. In this manuscript, the authors use a novel proximity labeling approach coupled to MS to identify substrates of TRIM2. Although the authors did not pinpoint the function of identified ubiquitylation sites on a novel TRIM2 substrate, I am enthusiastic about the manuscript due to high quality of experiments, data presentation, and clear conclusions. In addition to identifying 29 proteins as putative substrates that provide starting points for new studies, the authors validate TMEM 106B as a direct TRIM2 substrate and show that the two lysines in the cytosolic region are ubiquitylated. They also provide valuable structural information (using NMR and AF) on interaction surface between TMEM106 homodimerization surface and c-terminal part of TRIM2 and describe a new Zn-binding motif in TMEM106. Finally, the authors show that Zn binding by TMEM106 is important for lysosomal regulation.

Major points

Could the authors use published interaction datasets to check whether some of 29 putative substrates have been previously shown to interact or are even known substrates of TMEM109. This analysis could then be added in Figure 1. If the authors do not identify any previously known TMEM109 substrates this should be mentioned in text with possible explanations.

Did the authors in their proximity labeling or in in vitro ubiquitylation assays coupled to MS identify any Ub-linkage specific peptides that would help to understand linkage specificity of TRIM2. In any case, the authors should add a sentence about what is known about TRIM2 linkage specificity.

The authors should add in the main figure the effect of TRIM2 overexpression and knockdown on protein levels of TMEM106B. The statement that TMEM106 levels are not affected by TRIM2 ubiquitylation is currently not supported by sufficient data. If authors conclude that TMEM106B is not a proteasome substrate, but degraded through the

lysosome, then treatment with BafA should stabilize the protein. I think it would be important to perform this experiment as well as to determine the stability of TMEM106B in the absence and/or overexpression of TRIM2 using CHX assays.

For in vitro ubiquitylation assays, it would be important to state why the authors decided to use the particular E2 and whether they also tested any other. The fact that the authors observe only di-Ub may be because of sub-optimal.

2. Significance:

Significance (Required)

In this manuscript, the authors use a novel proximity labeling approach coupled to MS to identify substrates of TRIM2. Although the authors did not pinpoint the function of identified ubiquitylation sites on a novel TRIM2 substrate, I am enthusiastic about the manuscript due to high quality of experiments, data presentation, and clear conclusions.

3. How much time do you estimate the authors will need to complete the suggested revisions:

Estimated time to Complete Revisions (Required)

(Decision Recommendation)

Less than 1 month

4. Review Commons values the work of reviewers and encourages them to get credit for their work. Select 'Yes' below to register your reviewing activity at Web of Science Reviewer Recognition Service (formerly Publons); note that the content of your review will not be visible on Web of Science.

Yes

Review #2

1. Evidence, reproducibility and clarity:

Evidence, reproducibility and clarity (Required)

The study by Perez-Borrajero et al., offers significant insights into the interaction between TMEM106B and TRIM2, with a particular focus on the role of this interaction in lysosomal

regulation. The interaction is further explored by structural characterization, revealing that the zinc-binding motif in TMEM106B plays an important role in its dimerization which is essential for the lysosomal phenotype observed with TMEM106B overexpression. Although the details remain to be fully elucidated, the data contribute to the ongoing exploration of how ubiquitination modulates TMEM106B activity.

However, while the research identifies TMEM106B ubiquitination and its interaction with TRIM2, functional experiments are performed primarily in HEK293T cells, which may or may not have any relevance for the neurodegenerative diseases in which TMEM106B has been implicated.

1. Suggest performing some experiments in neurons or, at a minimum, neuronal cell lines (or microglia, or microglial cell lines). While it is true that neurons are not optimal for the screening experiments shown in Figure 1, once candidate interactions are identified, they can be verified in neurons. I think this work would be necessary for any higher-impact journal to consider the paper. Moreover, even among commonly used immortalized cell lines, HEK293Ts have very low TMEM106B expression (see Figure 2 in Brady et al, HMG, 2013). This matters because the authors are trying to extrapolate function from a cell line that has very low expression of the protein in question and is not implicated in any of the diseases in which TMEM106B plays an important role.
2. Supplemental Figure 4 is supposed to show that all the mutant TMEM106B constructs express similarly. However, there are several issues with this data. First, the C61S/C64S mutant is supposed to be different with respect to dimerization. However, the band is the same size as the other constructs, and with the "55kD" marker only, I cannot tell whether this band is the dimer or the monomer. The authors need to show the full blot, and they need to probe with an antibody against TMEM106B (not just the tag), which would also allow us to see how much over-expression is involved (because we would also see a band for the endogenous protein). Second, I do not think that the blot shown suggests similar levels of expression. Many of the bands appear over-saturated (which is why they may look "white" in the middle), making it nearly impossible to understand how much TMEM106B is being expressed. This point needs to be addressed.
3. Related to point #2, I am concerned that all the cellular experiments are performed in an over-expression context, because this can definitely cause artifacts. It is not that difficult, at this point, to edit TMEM106B to simply substitute the C61/64 residues, which would allow for comparisons of behavior at physiological levels. I don't think this must be done in a revision, but I really suggest doing this, in a more relevant cell line, to better understand impact.
4. What is necessary, however, is to show that the C61S/C64S mutants don't dimerize (or

reduce dimerization) in a cellular context. This can be accomplished by looking at cell lysates by immunoblotting, but I don't know why the blot in Supplemental Figure 4 doesn't seem to show dimer and monomer bands, which can be detected with a number of anti-TMEM106B antibodies (see Figure 1 in Chen-Plotkin et al, JNeurosci, 2012), or even with detection of tags (see Figure 1 in Brady et al, HMG, 2013). It could also be accomplished by mutating these residues, introducing the mutant into cells, and performing any number of interaction experiments (e.g. co-IP against different tags as in the Brady et al, HMG, 2013 paper) - this is different from the work shown in Figure 3 because I think they need to look at this in a cellular context.

5. Similarly, the authors argue that zinc binding is necessary for TMEM106B dimerization, but this is also never shown in a cellular context. This could be done with TMEM106B isolated from cells cultured with or without zinc supplementation to assess dimerization.

2. Significance:

Significance (Required)

The study by Perez-Borrajero et al., offers significant insights into the interaction between TMEM106B and TRIM2, with a particular focus on the role of this interaction in lysosomal regulation. The interaction is further explored by structural characterization, revealing that the zinc-binding motif in TMEM106B plays an important role in its dimerization which is essential for the lysosomal phenotype observed with TMEM106B overexpression. Although the details remain to be fully elucidated, the data contribute to the ongoing exploration of how ubiquitination modulates TMEM106B activity.

Please see Point 2 in my review -- this is the main issue I have with whether the evidence presented is rigorous and will be reproducible.

3. How much time do you estimate the authors will need to complete the suggested revisions:

Estimated time to Complete Revisions (Required)

(Decision Recommendation)

Between 1 and 3 months

4. Review Commons values the work of reviewers and encourages them to get credit for their work. Select 'Yes' below to register your reviewing activity at Web of Science Reviewer Recognition Service (formerly Publons); note that the content of your review will not be visible on Web of Science.

Yes

Review #3

1. Evidence, reproducibility and clarity:

Evidence, reproducibility and clarity (Required)

I have studied this paper, looking particularly at the NMR parts, because as someone who works on protein NMR I was felt to be a useful person to review this part in particular. In general, I find the results convincing, with only the structure determination being some small cause for concern.

The NMR spectra clearly show that TMEM106B is largely unstructured, but with a small structured region containing Cys61 and Cys64. The data presented in Gif 3a,b clearly show an interaction between TRIM2 and this region of the protein. Given that this is the only structured region of TMEM106B, the interaction with TRIM2 is not necessarily physiological, although it is certainly good circumstantial evidence for an interaction. I am not clear why residue K95 is highlighted in Fig 3a. This is the C-terminus of this construct, and does not look particularly different.

The evidence for crosslinking of the cysteines via a metal ion is good, and the experiments demonstrating that the metal is zinc are also fairly convincing. The SEC-MALS data are particularly clear, so overall the evidence for Zn-mediated dimerization is strong.

The only part of the NMR results that I would question is the structure (Fig 3g). The authors used distance restraints from Zn to cysteine S, and from S to S, as part of the structure calculation. These are extremely strong restraints, and place very tight limits on what the structure can do in this region. They derive from the assumption of Zn crosslinking, so do not form independent evidence for a Zn crosslink. It is therefore not surprising that there is a strong structural consistency in this region, and that the structures are very similar to AlphaFold predictions. I would therefore summarise this result as saying that the NMR data are consistent with the AlphaFold model, rather than that this is an NMR-determined structure, and I feel that it would be better to explain the result in this way. The Methods section provides little detail on how the structure calculation was done. If for example the NMR data (eg NOESY peaks) clearly indicate the presence of a dimer, which was made tighter by the addition of the Zn-S and S-S distances, then I would be much happier to call

this an NMR structure rather than a modelling exercise.

Fig 4a does indeed show that the cysteines are required for the formation of a dimeric structure, and form further good evidence for a Zn cross-link.

****Referee Cross-commenting****

Ref 2's comments about Supp Fig 4: I may be missing something, but how would you expect to see a dimer in an SDS-PAGE gel?

I like Ref 1's point about correlating the results with interactome databases - this would be useful.

2. Significance:

Significance (Required)

The NMR results provide convincing evidence for a structured dimeric region around residues 58-70, induced by Zn binding to the two pairs of cysteines. I find it surprising that such a stable and structured region can be formed by such a small set of interactions, so this is structurally a significant result. (Though I am perfectly happy to agree that compared to the rest of the paper it is of relatively low importance.)

I am less confident about the rest of the paper, but it feels a well constructed and supported study.

3. How much time do you estimate the authors will need to complete the suggested revisions:

Estimated time to Complete Revisions (Required)

(Decision Recommendation)

Less than 1 month

4. Review Commons values the work of reviewers and encourages them to get credit for their work. Select 'Yes' below to register your reviewing activity at Web of Science Reviewer Recognition Service (formerly Publons); note that the content of your review will not be visible on Web of Science.

Yes

Dear Prof. Hennig

Thank you for the submission of your research manuscript to our journal. I have now read your article, the referee reports and your point-by-point response. Taking all information into account, I agree that your manuscript might make an interesting contribution to our journal.

I therefore invite you to revise your manuscript along the lines suggested in your revision plan. We realize that it is difficult to revise to a specific deadline. In the interest of protecting the conceptual advance provided by the work, we recommend a revision within 3 months (September 23rd). Judging from the planned revisions, I feel that this timeframe appears adequate, but please let me know ahead of this time if you require more time to complete the revisions.

Acceptance of the manuscript will depend on a positive outcome of a second round of review. It is EMBO Reports policy to allow a single round of revision only and acceptance or rejection of the manuscript will therefore depend on the completeness of your responses included in the next, final version of the manuscript.

Your study will potentially be published in our Reports section. For short reports, the revised manuscript should not exceed 27,000 characters (including spaces but excluding materials & methods and references) and 5 main plus 5 expanded view figures. The results and discussion sections must further be combined, which will help to shorten the manuscript text by eliminating some redundancy that is inevitable when discussing the same experiments twice. For a normal article there are no length limitations, but it should have more than 5 main figures and the results and discussion sections must be separate. In both cases, the entire materials and methods must be included in the main manuscript file.

2) individual production quality figure files as .eps, .tif, .jpg (one file per figure).

Please download our Figure Preparation Guidelines (figure preparation pdf) from our Author Guidelines pages <https://www.embopress.org/page/journal/14693178/authorguide> for more info on how to prepare your figures.

4) a complete author checklist, which you can download from our author guidelines (). Please insert information in the checklist that is also reflected in the manuscript. The completed author checklist will also be part of the RPF.

5) Please note that all corresponding authors are required to supply an ORCID ID for their name upon submission of a revised manuscript (). Please find instructions on how to link your ORCID ID to your account in our manuscript tracking system in our Author guidelines
()

6) We replaced Supplementary Information with Expanded View (EV) Figures and Tables that are collapsible/expandable online. A maximum of 5 EV Figures can be typeset. EV Figures should be cited as 'Figure EV1, Figure EV2' etc... in the text and their respective legends should be included in the main text after the legends of regular figures.

7) Please include a dedicated "Data Availability" section at the end of the Methods (suggested wording: "The [structural coordinates | microarray | mass spectrometry] data from this publication have been deposited to the [name of the database] database [URL] and assigned the identifier [accession | permalink | hashtag]."). It is important that you include links that resolve directly to the dataset not just the database.

Additional information on source data and instruction on how to label the files are available

10) Figure legends and data quantification:

- the name of the statistical test used to generate error bars and P values,
 - the EXACT p-values,
 - the number (n) of independent experiments (please specify technical or biological replicates) underlying each data point,
 - the nature of the bars and error bars (s.d., s.e.m.)
- If the data are obtained from n {less than or equal to} 5, show the individual data points in addition to the SD or SEM.
- If the data are obtained from n {less than or equal to} 2, use scatter blots showing the individual data points.

11) Our journal encourages inclusion of *data citations in the reference list* to directly cite datasets that were re-used and obtained from public databases. Data citations in the article text are distinct from normal bibliographical citations and should directly link to the database records from which the data can be accessed. In the main text, data citations are formatted as follows: "Data ref: Smith et al, 2001" or "Data ref: NCBI Sequence Read Archive PRJNA342805, 2017". In the Reference list, data citations must be labeled with "[DATASET]". A data reference must provide the database name, accession number/identifiers and a resolvable link to the landing page from which the data can be accessed at the end of the reference. Further instructions are available at .

12) All Materials and Methods need to be described in the main text using our 'Structured Methods' format. According to this format, the Methods section includes a Reagents and Tools Table (listing key reagents, experimental models, software and relevant equipment and including their sources and relevant identifiers) followed by a Methods and Protocols section describing the methods, ideally using a step-by-step protocol format. The aim is to facilitate adoption of the methodologies across labs. Please download and fill our Reagents and Tools Table template (.docx), which you can find in our author guidelines:

13) As part of the EMBO publication's Transparent Editorial Process, EMBO Reports publishes online a Review Process File to accompany accepted manuscripts. This File will be published in conjunction with your paper and will include the referee reports, your point-by-point response and all pertinent correspondence relating to the manuscript.

Yours sincerely,

Reviewer #1 (Evidence, reproducibility and clarity (Required)):

I summarize the key strengths of the study and outline suggestions for key analysis / experiment that would strengthen the study below.

Ubiquitin E3 ligases provide specificity to the Ub system; however for most E3s substrates are poorly characterized. In this manuscript, the authors use a novel proximity labeling approach coupled to MS to identify substrates of TRIM2. Although the authors did not pinpoint the function of identified ubiquitylation sites on a novel TRIM2 substrate, I am enthusiastic about the manuscript due to high quality of experiments, data presentation, and clear conclusions. In addition to identifying 29 proteins as putative substrates that provide starting points for new studies, the authors validate TMEM 106B as a direct TRIM2 substrate and show that the two lysines in the cytosolic region are ubiquitylated. They also provide valuable structural information (using NMR and AF) on interaction surface between TMEM106 homodimerization surface and c-terminal part of TRIM2 and describe a new Zn-binding motif in TMEM106. Finally, the authors show that Zn binding by TMEM106 is important for lysosomal regulation.

Major points

Could the authors use published interaction datasets to check whether some of 29 putative substrates have been previously shown to interact or are even known substrates of TMEM109. This analysis could then be added in Figure 1. If the authors do not identify any previously known TMEM109 substrates this should be mentioned in text with possible explanations.

In the case of TRIM2 substrates, we now state in the text that none in the list of 29 substrates we identified had been previously reported. Reported substrates of TRIM2 include neurofilament light chain (NEFL) and Bcl-2-interacting mediator of cell death (BIM), and these were discovered through other (more traditional) methods. To our knowledge, this is the first study that uses proximity labeling to identify TRIM2 substrates, which could explain the new identifications. We have added a sentence on page 5 (last paragraph) to make that clearer.

If the reviewer meant known interactors of TMEM106B, we found some reported ones such as MAP6, TMEM106C, and GALC, but no published comprehensive interactome to see if TRIM2 appears on the list. We found one article identifying > 500 TMEM106B binding proteins, but we could not find the list (Takahashi et al., 2024). Of note, while TMEM106B is currently garnering attention due to links to neurodegeneration and viral infection, it remains relatively understudied.

Did the authors in their proximity labeling or in *in vitro* ubiquitylation assays coupled to MS identify any Ub-linkage specific peptides that would help to understand linkage specificity of TRIM2. In any case, the authors should add a sentence about what is known about TRIM2 linkage specificity.

The MS method used is not suitable to discern linkage types, because the trypsin digestion during processing of the samples removes the architectural information of the ubiquitin chains, and we detect only the signature Gly-Gly remnant at the modification site. We have now added a sentence in the Introduction (page 4, last paragraph) that states that there is currently no information in the literature regarding the linkage specificity of TRIM2.

To test linkage specificity, we used ubiquitin proteins lacking specific lysine residues and tested whether TRIM2 activity would be impaired in the presence of various mutants (e.g. Ubiquitin K63A or K48A). Using the same *in vitro* ubiquitination assays, we found no significant changes in the activity of TRIM2 among various mutants, and concluded that TRIM2 may use any available lysine, or perhaps require additional co-factors (e.g., specific E2s), to provide higher specificity in intracellular ubiquitination reactions. However, it remains a possibility that TRIM2 does not make specific ubiquitin chains but modifies more generally. We believe the question of linkage specificity of TRIM2 is outside the scope of this article.

The authors should add in the main figure the effect of TRIM2 overexpression and knockdown on protein

levels of TMEM106B. The statement that TMEM106 levels are not affected by TRIM2 ubiquitylation is currently not supported by sufficient data. If authors conclude that TMEM106B is not a proteasome substrate, but degraded through the lysosome, then treatment with BafA should stabilize the protein. I think it would be important to perform this experiment as well as to determine the stability of TMEM106B in the absence and/or overexpression of TRIM2 using CHX assays.

Since our manuscript was initially focused on the catalytic activity of TRIM2, we have compared TMEM106B levels in the presence of WT or catalytically dead versions of TRIM2 to assess the specific effect of ubiquitination using Western blots (WBs) (see Appendix Figure S2). If TRIM2 ubiquitination had a role to play in TMEM106B degradation, we would expect that TMEM106B levels increase when inactive TRIM2 is present, even at steady state (i.e., one time point). Comparing lanes 1 and 4/5, we see the opposite effect. If TRIM2 ubiquitination of TMEM106B had links to proteasome degradation, we would expect to see that TMEM106B accumulates in the presence of proteasome inhibitor and active TRIM2. Comparing lanes 1 and 6, we see also the opposite effect. We agree with the reviewer that CHX assays could help to establish the dynamics of TMEM106B degradation. However, this experiment requires that the protein is turned over relatively quickly (i.e. < 12 hours), and the assumption that TRIM2 affects this pathway. We partially address this point below while addressing Reviewer #2, but the evidence we have so far points to a non-degradative role of TRIM2 ubiquitination. In addition, mutation of the lysine residues in TMEM106B seems to have a destabilizing effect, which we can see better now with higher quality WBs post-revision (see response to Reviewer #2). Taken together, this would point to ubiquitination having a stabilizing rather than degradative role in the case of TMEM106B.

While we acknowledge that WBs are not generally quantitative, we believe that comparing active vs. inactive TRIM2 is better than looking at the presence vs. absence of TRIM2, because we cannot disentangle *bona fide* ubiquitination effects from artifacts from overexpression/knockdown. For instance, we found that merely including TRIM2 in the experiment was enough to reduce TMEM106B levels, independent of the ubiquitination activity, because co-transfection with two genes caused poorer protein expression.

TMEM106B regulates the formation, acidification, and trafficking of lysosomes (Rademakers et al., 2021; Schwenk et al., 2014). It was also shown to bind the v-ATPase accessory protein 1 (AP1) (Klein et al., 2017). Based on the literature, we think both BafA and TMEM106B affect lysosomal acidification/vacuolization, and we are unsure about the interpretation of potential experiments when using BafA given its link to the v-ATPase.

In our original text we state, "This is in line with previous observations that TMEM106B is likely turned over through the lysosomal pathway (Brady et al., 2013) and suggests that ubiquitination of TMEM106B by TRIM2 does not lead to proteasomal dependent degradation." We have now removed any reference to the lysosomal pathway to avoid overinterpretation and have kept our statement that our findings point to a proteasome-independent pathway for turnover of TMEM106B, consistent with the Brady *et al.* results. The changed sentence is on page 7 (end of last paragraph).

For *in vitro* ubiquitylation assays, it would be important to state why the authors decided to use the particular E2 and whether they also tested any other. The fact that the authors observe only di-Ub may be because of sub-optimal.

While initially studying TRIM2 activity, we tested a few promiscuous E2 conjugating enzymes that are well established in the literature to be functional with RING ligases like TRIM2, namely UBE2D1, UBE2D2, and UBE2D3. They were able to catalyze the TRIM2 autoubiquitination reaction, and we chose the most active one among these to test the activity towards TMEM106B. Of note, UBE2D1 was used in Esposito et al., 2022, *Nature Communications*, a structural study of TRIM2/3 ligases, and is highly similar to UBE2D3 (used here). We have now referred to this in the Methods section as an explanation for the choice of E2 (page 21, end of second paragraph).

We acknowledge that we do not know which of the ~ 40 E2 conjugating enzymes is best suited to work in combination with TRIM2 and TMEM106B *in vitro* or *in vivo*, and that this could explain the di-ubiquitination

observed in our reconstituted minimal system. Signaling cascades are quite flexible and context dependent, and different E2-E3 pairings could be preferred depending on the cell type and stimuli. While this question is interesting, we think it's outside the scope of the manuscript to establish the optimal E2 for the ubiquitination of TMEM106B by TRIM2.

Reviewer #1 (Significance (Required)):

In this manuscript, the authors use a novel proximity labeling approach coupled to MS to identify substrates of TRIM2. Although the authors did not pinpoint the function of identified ubiquitylation sites on a novel TRIM2 substrate, I am enthusiastic about the manuscript due to high quality of experiments, data presentation, and clear conclusions.

We thank the reviewer for the thorough analysis and excellent suggestions for improvement and future directions.

Reviewer #2 (Evidence, reproducibility and clarity (Required)):

The study by Perez-Borrajero et al., offers significant insights into the interaction between TMEM106B and TRIM2, with a particular focus on the role of this interaction in lysosomal regulation. The interaction is further explored by structural characterization, revealing that the zinc-binding motif in TMEM106B plays an important role in its dimerization which is essential for the lysosomal phenotype observed with TMEM106B overexpression. Although the details remain to be fully elucidated, the data contribute to the ongoing exploration of how ubiquitination modulates TMEM106B activity.

However, while the research identifies TMEM106B ubiquitination and its interaction with TRIM2, functional experiments are performed primarily in HEK293T cells, which may or may not have any relevance for the neurodegenerative diseases in which TMEM106B has been implicated.

1. Suggest performing some experiments in neurons or, at a minimum, neuronal cell lines (or microglia, or microglial cell lines). While it is true that neurons are not optimal for the screening experiments shown in Figure 1, once candidate interactions are identified, they can be verified in neurons. I think this work would be necessary for any higher-impact journal to consider the paper. Moreover, even among commonly used immortalized cell lines, HEK293Ts have very low TMEM106B expression (see Figure 2 in Brady et al, HMG, 2013). This matters because the authors are trying to extrapolate function from a cell line that has very low expression of the protein in question and is not implicated in any of the diseases in which TMEM106B plays an important role.

We performed experiments in mouse embryonic stem cell (mESC) derived neurons at day 12 of differentiation when they are considered glutamatergic. Specifically, we knocked out the TRIM2 and closely related TRIM3 full-length proteins using CRISPR-Cas9 and performed differential quantitative proteomics to study changes in protein abundance in the absence of TRIM2/3. The results are shown in the figure below:

Figure for referee with unpublished data and its description has been removed upon request by the authors.

We found i) that TMEM106B levels increase upon differentiation from the mESC stage (day 0) to neurons (day 12), as expected given its prominent role in neurobiology, and ii) that TMEM106B levels do not consistently nor significantly change upon removal of TRIM2/3.

We chose not to include these data in the manuscript because we felt the quality of the experiment was inadequate and the results tangential. We did not obtain CRISPR clones with sufficient quality, and the fact that we were unable to knock out TRIM2 on its own also complicates the interpretation of the data because the regulatory interplay of the TRIM2 and TRIM3 proteins is unclear. However, these results in neurons would support our conclusion in the manuscript that TRIM2 does not target TMEM106B for degradation, as its levels do not significantly change with TRIM2/3 removal. We believe our study lays the foundation for more detailed investigations into how Zn-mediated dimerization can influence the function of TMEM106B in neurons or other cell types and agree with the reviewer that future work should focus on fine control of TMEM106B expression in systems suitable to study neurodegeneration or other cellular processes.

2. Supplemental Figure 4 is supposed to show that all the mutant TMEM106B constructs express

similarly. However, there are several issues with this data. First, the C61S/C64S mutant is supposed to be different with respect to dimerization. However, the band is the same size as the other constructs, and with the "55kD" marker only, I cannot tell whether this band is the dimer or the monomer. The authors need to show the full blot, and they need to probe with an antibody against TMEM106B (not just the tag), which would also allow us to see how much over-expression is involved (because we would also see a band for the endogenous protein). Second, I do not think that the blot shown suggests similar levels of expression. Many of the bands appear over-saturated (which is why they may look "white" in the middle), making it nearly impossible to understand how much TMEM106B is being expressed. This point needs to be addressed.

We agree with the reviewer that the blot shown was oversaturated and have re-done it with diluted samples to better discern differences in expression levels. The affected figure panel (Appendix Figure S4A) has been replaced with a higher quality WB.

The mutants of TMEM106B appear roughly the same size due to the use of a denaturing SDS-PAGE, which is unable to discern oligomerization states because the TMEM106B protein becomes unfolded, loses any co-factor (e.g. Zn) and becomes negatively charged with the SDS detergent. Only the mutation of both lysine residues to alanine causes a slight shift, which is indicated in the text/figures and most likely arises from the change in hydrophobic properties.

Below is the data we collected with variations of the Western blot:

We now see that the K3A/K14A mutant has somewhat lower expression levels (left WB), which could hint at ubiquitination having a stabilizing role. Importantly, our conclusions are not affected by this result, since the lysosomal phenotype observed with the K3A/K14A mutant is slightly more pronounced than that observed with WT TMEM106B, and it remains true that the variants express at comparable levels.

We see a similar pattern using an antibody directly against TMEM106B (middle WB), indicating that endogenous TMEM106B does not interfere with the experiment. Endogenous TMEM106B is likely the protein band with slightly lower molecular weight, present in low amounts in all samples.

The WB on the right is derived from Native-PAGE with the same samples. Consistent with the conclusions of our study, the C61S/C64S mutations cause a defect in TMEM106B which is appreciable also when probing the structural integrity of the protein produced in HEK 293-T cells, and consistent with the interactome experiment added to the manuscript (see point 5 below). The samples of this mutant (last two lanes) show a heterogeneous rather than defined migration pattern, which would be consistent with the Zinc-binding site being key to the structural integrity of the protein and dimerization, and thus

changing the appearance on a Native-PAGE. Of note, we checked whether the C61S/C64S samples had degraded, which could also explain the pattern observed, and this was not the case, as we performed the SDS-PAGE WB again and obtained the same results. Moreover, our *in vitro* data show convincingly that this mutant is monomeric (NMR, SEC-MALS).

3. Related to point #2, I am concerned that all the cellular experiments are performed in an over-expression context, because this can definitely cause artifacts. It is not that difficult, at this point, to edit TMEM106B to simply substitute the C61/64 residues, which would allow for comparisons of behavior at physiological levels. I don't think this must be done in a revision, but I really suggest doing this, in a more relevant cell line, to better understand impact.

We agree with the reviewer's point that a better understanding of TMEM106B function requires a more physiologically relevant system, as well as editing tools to finely control its expression. Nevertheless, the lysosomal enlargement phenotype (i.e. vacuolization) resulting from TMEM106B overexpression observed in HEK 293 cells is equivalent to that recently found in the brains of transgenic mice (Perneel et al., 2025), and has become well established in the literature, with multiple articles making identical observations across diverse cellular systems (Brady et al., 2013; Chen-Plotkin et al., 2012; Feng et al., 2020). No studies had previously examined the mostly disordered, but clearly regulatory cytosolic region of TMEM106B, and the use of the lysosomal phenotype to test its functionality has proven to be highly reproducible. In addition, although TMEM106B has garnered attention mainly because of its link to neurodegeneration, it is also highly expressed in muscle, endocrine, and male reproductive tissues (Uhlén et al., 2015), and it is present naturally, albeit with lower expression, in HEK 293 cells.

4. What is necessary, however, is to show that the C61S/C64S mutants don't dimerize (or reduce dimerization) in a cellular context. This can be accomplished by looking at cell lysates by immunoblotting, but I don't know why the blot in Supplemental Figure 4 doesn't seem to show dimer and monomer bands, which can be detected with a number of anti-TMEM106B antibodies (see Figure 1 in Chen-Plotkin et al, JNeurosci, 2012), or even with detection of tags (see Figure 1 in Brady et al, HMG, 2013). It could also be accomplished by mutating these residues, introducing the mutant into cells, and performing any number of interaction experiments (e.g. co-IP against different tags as in the Brady et al, HMG, 2013 paper) - this is different from the work shown in Figure 3 because I think they need to look at this in a cellular context.

5. Similarly, the authors argue that zinc binding is necessary for TMEM106B dimerization, but this is also never shown in a cellular context. This could be done with TMEM106B isolated from cells cultured with or without zinc supplementation to assess dimerization.

We have addressed reviewer's point 4 and point 5 with the following experiments:

i) co-IP of TMEM106B variants (WT, C61S/C64S, and K3A/K14A) to check the effect of dimerization/Zn binding on protein-protein interactions. The flag-tagged TMEM106B variants were transfected into HEK 293-T cells, and protein interactions identified with quantitative MS after pull-downs. The results clearly show that a number of protein-protein interactions are no longer possible with the C61S/C64S mutant of TMEM106B relative to WT, while the K3A/K14A mutant did not show many differences. Specifically, 8 protein interactions were disrupted upon mutation of the cysteine but not the lysine residues. Although this does not directly show zinc-binding or dimerization in cells, it is consistent with the cysteine residues having a key role in TMEM106B regulation in cells at the molecular level, in combination with the confocal microscopy, structural data, and Native-PAGE. We have now added these data in the main text (page 11, 2nd paragraph, page 13, first paragraph and Figure 5) and Appendix.

ii) Native-PAGE WB (see point 2 above). We opted for not including the native page into our manuscript as the native page only shows a smear and not a clear monomeric band. However, the dimeric band is not clearly discernable. We believe that we have shown convincingly with our *in vitro* experiments that the cysteine mutant of TMEM106B is monomeric and that Zn is needed for dimerization. This is also supported by the experimental NMR structure.

Reviewer #2 (Significance (Required)):

The study by Perez-Borrajero et al., offers significant insights into the interaction between TMEM106B and TRIM2, with a particular focus on the role of this interaction in lysosomal regulation. The interaction is further explored by structural characterization, revealing that the zinc-binding motif in TMEM106B plays an important role in its dimerization which is essential for the lysosomal phenotype observed with TMEM106B overexpression. Although the details remain to be fully elucidated, the data contribute to the ongoing exploration of how ubiquitination modulates TMEM106B activity.

Please see Point 2 in my review -- this is the main issue I have with whether the evidence presented is rigorous and will be reproducible.

We thank the reviewer for the careful analysis of our manuscript and great suggestions for testing the intracellular relevance of the cysteine-mediated regulatory region.

Reviewer #3 (Evidence, reproducibility and clarity (Required)):

I have studied this paper, looking particularly at the NMR parts, because as someone who works on protein NMR I was felt to be a useful person to review this part in particular. In general, I find the results convincing, with only the structure determination being some small cause for concern.

The NMR spectra clearly show that TMEM106B is largely unstructured, but with a small structured region containing Cys61 and Cys64. The data presented in Gif 3a,b clearly show an interaction between TRIM2 and this region of the protein. Given that this is the only structured region of TMEM106B, the interaction with TRIM2 is not necessarily physiological, although it is certainly good circumstantial evidence for an interaction. I am not clear why residue K95 is highlighted in Fig 3a. This is the C-terminus of this construct, and does not look particularly different.

We chose to highlight differences between interacting residues like C61/64 and non-interacting residues like K95. As the reviewer pointed out, K95 does not look particularly different, and this peak was meant to serve as a "control" (i.e., relatively unperturbed). The way it was written in the text was unclear and we have now corrected this (figure legend to Figure 3, panel A).

The evidence for crosslinking of the cysteines via a metal ion is good, and the experiments demonstrating that the metal is zinc are also fairly convincing. The SEC-MALS data are particularly clear, so overall the evidence for Zn-mediated dimerization is strong.

The only part of the NMR results that I would question is the structure (Fig 3g). The authors used distance restraints from Zn to cysteine S, and from S to S, as part of the structure calculation. These are extremely strong restraints, and place very tight limits on what the structure can do in this region. They derive from the assumption of Zn crosslinking, so do not form independent evidence for a Zn crosslink. It is therefore not surprising that there is a strong structural consistency in this region, and that the structures are very similar to AlphaFold predictions. I would therefore summarise this result as saying that the NMR data are consistent with the AlphaFold model, rather than that this is an NMR-determined structure, and I feel that it would be better to explain the result in this way. The Methods section provides little detail on how the structure calculation was done. If for example the NMR data (eg NOESY peaks) clearly indicate the presence of a dimer, which was made tighter by the addition of the Zn-S and S-S distances, then I would be much happier to call this an NMR structure rather than a modelling exercise.

Fig 4a does indeed show that the cysteines are required for the formation of a dimeric structure, and form further good evidence for a Zn cross-link.

The reviewer asks to what extent the NMR structure obtained relies on strong Zn distance restraints and whether these restraints are the reason for the agreement between the NMR structure and the AlphaFold prediction.

We are confident that TMEM106B is bound by Zn due to several lines of evidence. First, the supplementation of Zn to a sample of WT TMEM106B¹⁻⁹⁵ containing both monomeric and dimeric species causes a shift to the purely dimeric species, indicating binding (Figure 3F). Second, mutation of the cysteine residues involved in coordination renders the protein monomeric (Figure 3C). Third, the beta carbon chemical shifts of the cysteine residues involved in coordination are consistent with metal binding, and not with the typical values for either a reduced (SH) or oxidized (S-S) state (Appendix Figure S3E). Fourth, the experiments with EDTA show an effect in the structure consistent with a shift to the monomeric variant (NMR and SEC-MALS data). Finally, we present below new 1D spectra of the unlabeled TMEM106B peptide used for the structure calculation before and after the addition of Zn. Only in the presence of Zn can one see the peak dispersion from the structured region.

While we do not have a crystal structure, we are not aware of reported Zn atoms crosslinking to S, but instead believe this is a non-covalent Zn binding site, whereby the S atoms provide free electrons that

allow Zn to be coordinated stably in a tetrahedral arrangement, as often found in Zn-binding proteins. This is the most likely scenario and is routinely used in NMR structure calculations once Zn binding and the oligomerization state of the protein has been established (Neuhaus, 2022).

To drive the folding of the backbone, the NMR structure relies on 145 additional experimentally determined intermolecular distance restraints derived from NOESY spectra and obtained with the automatic `noe_assign` function of CYANA. The figure below shows a part of the NOESY with unambiguously and manually assigned intermolecular NOESY cross peaks as examples.

These NOE contacts identify the intermolecular β -sheet that arise from the dimer and contribute to folding. We have now added this information in the Methods (page 23, last paragraph) and Appendix sections for greater clarity on the structure calculation. To test the contribution of strong Zn restraints, we have performed the structure calculation without these and compared the results in the figure below.

The superposition of the lowest-energy structures from calculations with (cyan) or without (red) Zn and Zn-S restraints is shown. The backbone RMSD is 1.1 Å for the structured regions (residues 57-71), and the dimeric interface is present in both.

Therefore, the applied Zn restraints in the NMR structure deposited, while contributing to the proper geometry and orientation of the coordination site (and the addition of the Zn ion itself), are not required for obtaining the experimental NMR structure that AlphaFold predicts.

****Referee Cross-commenting****

Ref 2's comments about Supp Fig 4: I may be missing something, but how would you expect to see a dimer in an SDS-PAGE gel?

We would not expect to see a dimer on an SDS-PAGE and have now mentioned this in the corresponding section. We thank the reviewer for supporting our answer.

I like Ref 1's point about correlating the results with interactome databases - this would be useful.

We have addressed this in the corresponding section.

Reviewer #3 (Significance (Required)):

The NMR results provide convincing evidence for a structured dimeric region around residues 58-70, induced by Zn binding to the two pairs of cysteines. I find it surprising that such a stable and structured region can be formed by such a small set of interactions, so this is structurally a significant result. (Though I am perfectly happy to agree that compared to the rest of the paper it is of relatively low importance.)

I am less confident about the rest of the paper, but it feels a well constructed and supported study.

We thank the reviewer for the close look at the biophysical data and suggestions for improvement.

Dear Janosch,

Thank you once more for the submission of your revised manuscript to EMBO reports and for your feedback on the remaining concerns from former referee #2. As discussed, please address the remaining concerns in the manuscript text and discussion and please also provide a point-by-point response.

From the editorial side, there are also a few things that we need before we can proceed with the official acceptance of your study.

- Your article will be published in our Reports section, which would require a combined Results and Discussion section.
- Please provide up to 5 keywords on the title page.
- Regarding the Author Contributions, we now use CRediT to specify the contributions of each author in the journal submission system. Therefore, please remove the Author Contributions from the manuscript file and make sure that the author contributions in our online manuscript tracking system are correct and up-to-date. The information you specified in the system will be automatically retrieved and typeset into the article. You can enter additional information in the free text box provided, if you wish. See also our guide to authors <https://www.embopress.org/page/journal/14693178/authorguide#authorshipguidelines>.
- References: et al needs to be used after 10 author names; DOIs should only be used for preprints and datasets that have not been published yet
- Evans et al, 2022 is a preprint. It should therefore be cited as follows:
In the text as (preprint: Evans et al, 2022).
In the reference list it needs the tag [PREPRINT] at the end of the reference.
- Please provide a callout for "Appendix Table S1" in the text.
- The paragraph "Supplemental Material" should be removed from the manuscript. The titles listed there should be added to the "Description" tab in the .xls file itself. Or at least the file name (Dataset EV#, Table EV1) needs to be there.
- Dataset EV4 is not really a dataset. It should be uploaded as Table EV1 and all references, legends, and file titles be updated accordingly incl in the Author Checklist and the Reagents and Tools table.
- Data availability section: Please provide URLs that resolve directly to the datasets. Currently this is missing for 9GI8 and PXD055297.
- Appendix Figure S4B: please define the scale bar size only in the legend, not in the figure panel.
- Please move the figure legends to the end of the manuscript with the header "Figure Legends".
- Materials and Methods should be Methods
- Please remove the "Instructions" paragraph from the Reagents and Tools table.
- During our routine image checks, we noticed that the images within the Appendix file appear pixelated under analysis. This is a common result of converting original 16-bit TIFF images to RGB format for publication, and while not a cause for concern, it can sometimes give the impression of image alteration to critical readers.

To resolve this please upload the Appendix file at a higher resolution.

- Our team of data editors has checked the figure legends and asked you to address the following points:
 - 1) Please indicate the statistical test used for data analysis in the legends of figures 1C, 5B, S1 B, S4C
 - 2) Please note that information related to n is missing in the legends of figures 1C, 4B, S4C
 - 3) Please note that the error bars are not defined in the legend of figure 4B
- Finally, EMBO Reports papers are accompanied online by
 - A) a short (1-2 sentences) summary of the findings and their significance,
 - B) 2-3 bullet points highlighting key results and
 - C) a schematic summary figure that provides a sketch of the major findings (not a data image).Please provide the summary figure as a separate file in PNG or JPG format at a size of 550x300-600 pixels (width x height). Please note that the size is rather small and that text needs to be readable at the final size. Please send us this information

along with the revised manuscript.

With kind regards,

Martina

=====

Referee #2:

The authors have addressed several of my concerns, but some issues remain.

1. I still think that the main scientific audience interested in TMEM106B function is in relation to its role in neurodegeneration and/or SARS-COV2 and, as stated earlier, HEK293 cells are very low TMEM106B expressors. My worry is that if there is no confirmation of TRIM2 ubiquitination/regulation of TMEM106B function in any relevant cell type, this limits the impact of this work substantially. I agree that the mouse embryonic stem cell derived neuron data is tangential to the current work, but I think that does not negate my original point. In the end, this question of what is necessary for the audience of EMBO Reports is best decided by the editors of EMBO reports!

2. The new immunoblot in S4 is slightly helpful as are the other blots in the author response. It is puzzling that they cannot detect a dimer band for TMEM106B, as this has been widely reported in the literature, by multiple labs (see PMID 23136129, 22895706, 37745346, 39709600, among many other examples). I disagree with the third reviewer that no dimers can be seen on SDS-page gel, since that depends on the nature of the dimer, and how the sample was treated - it is widely appreciated in the field that immunoblotting TMEM106B is best done in the absence of heating/boiling and BME. Their results may depend on how they have handled their sample and what anti-TMEM106B antibody was used. Because so much of their interpretation hinges on dimerization, they should blot with established antibodies that have been previously reported to detect TMEM106B dimer under the non-reducing, non-heat-treated conditions, as I still cannot understand why they are seeing a 55kD band when most labs (including the 4 labs whose papers are listed above) detect bands for TMEM106B at ~37kD and 75kD, and I also find myself wondering what the lower band in their anti-FLAG blot (in the author response, cut off of S4) represents.

Referee #2:

The authors have addressed several of my concerns, but some issues remain.

1. I still think that the main scientific audience interested in TMEM106B function is in relation to its role in neurodegeneration and/or SARS-COV2 and, as stated earlier, HEK293 cells are very low TMEM106B expressors. My worry is that if there is no confirmation of TRIM2 ubiquitination/regulation of TMEM106B function in any relevant cell type, this limits the impact of this work substantially. I agree that the mouse embryonic stem cell derived neuron data is tangential to the current work, but I think that does not negate my original point. In the end, this question of what is necessary for the audience of EMBO Reports is best decided by the editors of EMBO reports!

2. The new immunoblot in S4 is slightly helpful as are the other blots in the author response. It is puzzling that they cannot detect a dimer band for TMEM106B, as this has been widely reported in the literature, by multiple labs (see PMID 23136129, 22895706, 37745346, 39709600, among many other examples). I disagree with the third reviewer that no dimers can be seen on SDS-page gel, since that depends on the nature of the dimer, and how the sample was treated - it is widely appreciated in the field that immunoblotting TMEM106B is best done in the absence of heating/boiling and BME. Their results may depend on how they have handled their sample and what anti-TMEM106B antibody was used. Because so much of their interpretation hinges on dimerization, they should blot with established antibodies that have been previously reported to detect TMEM106B dimer under the non-reducing, non-heat-treated conditions, as I still cannot understand why they are seeing a 55kD band when

most labs (including the 4 labs whose papers are listed above) detect bands for TMEM106B at ~37kD and 75kD, and I also find myself wondering what the lower band in their anti-FLAG blot (in the author response, cut off of S4) represents.

Rev_Com_number: RC-2025-02957

New_manu_number: EMBOR-2025-62088V2

Corr_author: Hennig

Title: TRIM2 E3 ligase substrate discovery reveals zinc regulation of TMEM106B in the endolysosomal pathway

Reviewer #2:

1. I still think that the main scientific audience interested in TMEM106B function is in relation to its role in neurodegeneration and/or SARS-COV2 and, as stated earlier, HEK293 cells are very low TMEM106B expressors. My worry is that if there is no confirmation of TRIM2 ubiquitination/regulation of TMEM106B function in any relevant cell type, this limits the impact of this work substantially. I agree that the mouse embryonic stem cell derived neuron data is tangential to the current work, but I think that does not negate my original point. In the end, this question of what is necessary for the audience of EMBO Reports is best decided by the editors of EMBO reports!

We would like to emphasize again that HEK293 cells have been used previously (see the PMIDs below for example) as a model system to understand TMEM106B function. TMEM106B and TRIM2 are both overexpressed in mouse embryonic stem cell differentiating into neurons. In our previous rebuttal letter, we already showed that TRIM2 is highly expressed in neuronal stem cells, data the reviewer agrees does not need to be included in the manuscript and TMEM106B overexpression in such cells has been well documented. Although we cannot be certain, there is no conceptual reason to assume TRIM2 would not ubiquitinate TMEM106B in neuronal contexts.

As is standard in mechanistic cell biology, model systems are often used when more physiologically relevant systems are technically intractable. This is particularly true for neuronal stem cells, which are extremely challenging to transfect. The articles the reviewer mentions in the second point below similarly rely on HEK293 cells or brain tissue lysates rather than alternative cell lines.

More importantly, the reviewer reduces our manuscript to ubiquitination and overlooks the other exciting novelties of our study: we identify a previously unrecognized Zn-dependent structural dimerization motif in the N-terminal cytosolic region of TMEM106B previously thought to be intrinsically disordered. This motif features tetrahedral Zn coordination by cysteine residues, a conceptually exciting finding and only slightly reminiscent of the Zn-hook motif (Hopfner et al., Nature, 2002). Discovering a new structural motif of this nature is in itself a significant contribution and nowadays rarely happens. Furthermore, we demonstrate that perturbing this motif through cysteine mutations, thereby abolishing dimerization profoundly disrupts lysosome assembly. These insights extend far beyond the specific question of TRIM2-mediated ubiquitination.

Nevertheless, we have added a small paragraph in the results and discussion section (page 10, end of 2nd paragraph), which describes the limitation of the study and that it remains to be seen whether TRIM2 ubiquitinates TMEM106B also in neuronal cells.

2. The new immunoblot in S4 is slightly helpful as are the other blots in the author response. It is puzzling that they cannot detect a dimer band for TMEM106B, as this has been widely reported in the literature, by multiple labs (see PMID 23136129, 22895706, 37745346, 39709600, among many other examples). I disagree with the third reviewer that no dimers can be seen on SDS-page gel, since that depends on the nature of the dimer, and how the sample was treated - it is widely appreciated in the field that immunoblotting TMEM106B is best done in the absence of heating/boiling and BME. Their results may depend on how they have handled their sample and what anti-TMEM106B antibody was used. Because so much of their interpretation hinges on dimerization, they should blot with established antibodies that have been previously reported to detect TMEM106B dimer under the non-reducing, non-heat-treated conditions, as I still cannot understand why they are seeing a 55kD band when most labs (including the 4 labs whose papers are listed above) detect bands for TMEM106B at ~37kD and 75kD, and I also find myself wondering what the lower band in their anti-FLAG blot (in the author response, cut off of S4) represents.

The reviewer remains puzzled by our immunoblot results and questions why we do not detect a dimer band for TMEM106B under denaturing conditions, citing several publications as evidence. We have carefully revisited each of the referenced studies, and their interpretation is not as uniform as the reviewer suggests.

- PMID 22895706: This study reports elevated apparent molecular weights (40 and 75 kDa) attributed to glycosylation; the dimer band disappears upon heat treatment, exactly consistent with our approach. (the authors do not state whether they use 1xFlag or 3xFlag).

- PMID 23136129: Only a monomer band is shown; the dimer is mentioned (of note: mentioned to be of 86 kDa of size) but not shown, and its apparent molecular weight differs substantially. 3xFlag is used.
- PMID 37745346: A single gel/blot: Under mild conditions a dimer band is visible. Apparent weight is considerably below 75 kDa.

- PMID 39709600: Dimer detection is inconsistent and appears only under mild treatment conditions; band sizes vary and are considerably higher than the 40 and 75 kDa the reviewer cites (rather 50 and 100).

These studies, together with ours, highlight that TMEM106B dimers are sensitive to sample handling and posttranslational modifications, including tag effects. We use a 3xFLAG tag, which is well known to alter apparent migration. The variability in reported molecular weights is fully compatible with differences in experimental conditions.

Crucially, in the revised manuscript and response letter we have already included:

- a co-IP confirming the functional relevance of the Zn-dependent motif in cells, and
- a native gel from cells demonstrating the disappearance of the dimer band in cysteine mutants.

This directly addresses the reviewer's original concern about demonstrating the loss of dimerization in cells.

The reviewer's current insistence therefore seems less about scientific substance and more about a difference in methodological preference. The fundamental point, that the Zn-dependent motif we have identified mediates TMEM106B dimerization is supported both in vitro and in cells, and our data are entirely consistent with the published literature when differences in experimental conditions are considered.

Again, we would like to emphasize that dimerization was confirmed for TMEM106B before, but it was not known what causes dimerization. We have provided ample evidence in vitro and in cells, that this novel Zn-dependent structural motif is responsible for this functionally relevant dimerization and have provided sufficient proof and controls.

We have extended the introduction and discussions of our revised manuscript to address this point.

We hope that this and our rebuttal clarifies why we view the remaining points raised by reviewer 2 as disproportionate to the actual scientific issues and not reflective of the overall strength of our findings. We remain confident that our study provides novel mechanistic insight of clear interest to the readership of EMBO Reports. We thank the reviewer for the careful analysis of our manuscript and great suggestions for testing the intracellular relevance of the cysteine-mediated regulatory region.

Prof. Janosch Hennig
Universität Bayreuth
Department of Biochemistry IV
Universitätsstrasse 31
Bayreuth 95447
Germany

Dear Prof. Hennig,

I am very pleased to accept your manuscript for publication in the next available issue of EMBO reports. Thank you for your contribution to our journal.

Kind regards,

Rev_Com_number: RC-2025-02957

New_manu_number: EMBOR-2025-62088V3

Corr_author: Hennig

Title: TRIM2 E3 ligase substrate discovery reveals zinc regulation of TMEM106B in the endolysosomal pathway